# BEYOND CORRELATION: THE IMPACT OF HUMAN UNCERTAINTY IN MEASURING THE EFFECTIVENESS OF AUTOMATIC EVALUATION AND LLM-AS-A-JUDGE

**Aparna Elangovan**[1]    **Lei Xu**[1†]    **Jongwoo Ko**[2†]    **Mahsa Elyasi**[1†]
**Ling Liu**[1]    **Sravan Bodapati**[1]    **Dan Roth**[1]

[1]Amazon        [2]KAIST AI        [†]Equal contributions

## ABSTRACT

The effectiveness of automatic evaluation of generative models is typically measured by comparing the labels generated via automation with human labels using correlation metrics. However, metrics like Krippendorff's $\alpha$ and Randolph's $\kappa$ were originally designed to measure the reliability of human labeling, thus make assumptions about typical human labeling behavior, and these assumptions may not be applicable to machine generated labels. In this paper, we show how *relying on a single aggregate correlation score* can obscure fundamental differences between human labels and those from automatic evaluation, including LLM-as-a-Judge. Specifically, we demonstrate that when the proportion of samples with variation or uncertainty in human assigned labels is relatively high, machine labels (generated by automatic evaluation methods) may superficially appear to have similar or better correlation with the human majority label compared to the human-to-human (HH) correlation. This can create the illusion that labels from automatic evaluation approximates the human majority label. However, as the proportion of samples with consistent human labels increases, the correlation between machine and human labels fall well below HH correlation. Based on these findings, we first propose *stratifying data by human label uncertainty* to provide a more robust analysis of automatic evaluation performance. Second, recognizing that uncertainty and variation are inherent in perception-based human evaluations, such as those involving attitudes or preferences, we introduce a new metric - *binned Jensen-Shannon Divergence for perception* for such scenarios to better measure the effectiveness of automatic evaluations. Third, we present visualization techniques – *perception charts*, to contextualize correlation measures appropriately and to show the strengths and limitations of automatic evaluation. We have open-sourced our analysis and visualization tools at https://github.com/amazon-science/BeyondCorrelation.

## 1 INTRODUCTION

Despite the increasing importance of automatic evaluation for generative large language models (LLMs), measuring and interpreting the effectiveness of these methods remains a challenge. Automatic evaluation methods include $N$-gram-based methods such as Rouge (Lin, 2004) and BERTScore (Zhang* et al., 2020), task-specific model-based methods such as AlignScore (Zha et al., 2023) and Prometheus (Kim et al., 2024), and LLM-as-a-Judge-based methods (LJ) where a general purpose LLM, using prompts, is instructed to perform evaluation. The use of automatic evaluation is tied to the key question – *How can the efficacy of automatic evaluation methods be measured in comparison to human evaluation?* A widely accepted practice is to use agreement or rank correlation metrics such as Krippendorff's-$\alpha$ (Krippendorff, 2011), Cohen's-$\kappa$ (Cohen, 1960; McHugh, 2012), Spearman's $\rho$, and Kendall's $\tau$-b (Fabbri et al., 2021; Deutsch et al., 2022; Chiang & Lee, 2023; Liu et al., 2023b), where a higher correlation between *machine labels* (generated by automatic evaluation methods) and *human labels* (gathered during human evaluation) implies a more effective automatic evaluation. Variation or uncertainty in human labels (Tversky & Kahneman, 1974) is almost impossible to avoid when humans are involved (Kahneman et al., 2021). In this

paper, we highlight how comparing the overall correlation score between human labels and machine labels can obscure key discrepancies in machine labels. Therefore, we propose that measurements be stratified by label uncertainty.

The impact of human label uncertainty on automatic evaluation, therefore, leads to a follow-on question – how then to effectively account for human uncertainty in automatic evaluation. Not all types of uncertainties are equal (Kahneman & Tversky, 1982), as they can range from random human errors, ambiguity in tasks, and a by-product of perception-based evaluation (Elangovan et al., 2024). Some human evaluation tasks naturally have higher variance than others. For instance, when evaluating aspects such as "*How readable is the given content?*", the answer relies on human perception and is likely to vary from person to person. In contrast, evaluating a natural language inference (NLI) task, such as "*Can you infer the given sentence from a given text?*", is less likely to show variations. Acceptable variation in labels, therefore, depends on the task, as *variations or uncertainties in human labels can sometimes be a feature rather than a bug* in human evaluation. Thus, we explore how to effectively measure and interpret reliability of automatic evaluation while taking human uncertainty in to consideration. We ask the following research questions:

- **RQ1:** How does uncertainty in human labels impact correlation metrics when we measure the efficacy of automatic evaluation methods? (*See detailed analysis on stratifying data by human uncertainty in* **Sec. 3.1** )
- **RQ2:** How can we measure human-machine agreement that accounts for human uncertainty as a result of variation in human perception? (*See Binned Jensen-Shannon divergence in* **Sec. 3.1** )
- **RQ3:** How can we visualize the underlying data to draw meaningful insights when we compare the results from automatic and human evaluations? (*See Perception charts in* **Sec. 3.3** )

## 2 BACKGROUND ON CORRELATION MEASURES

### 2.1 CORRELATION MEASUREMENTS AND ASSUMPTIONS

Correlation measures have traditionally been used to measure the inter-rater agreement (IRA) of human-generated labels, where there is no ground truth available, to measure human-to-human (HH) correlation. There are 2 broad categories of correlation measures, **(a)** non-chance-adjusted indices and **(b)** chance-adjusted metrics (Xinshu Zhao & Deng, 2013). Chance-adjusted measures take into account a scenario where raters can choose the same label by chance even when they do not know the correct answer. Chance-adjusted measures, such as Krippendorff's-$\alpha$ are generally more popular IRA measures than non-chance-adjusted metrics such as percentage agreement. The key difference between various chance-adjusted IRA measures is how chance agreement is computed (Meyer et al., 2014; Xinshu Zhao & Deng, 2013). For instance, Randolph's-$\kappa$ (Randolph, 2005) assumes that all labels are equally likely and assigns a probability of $1/k$ for chance agreement, where $k$ is the number of possible label choices. On the other hand, Fleiss's $\kappa$ (Fleiss, 1971) and Krippendorff's $\alpha$ calculate chance agreement empirically, based on the actual human labels of the study. These measurements can be useful in cases where certain labels (e.g., 3 on a Likert scale of 1-5) are more likely to be chosen when humans are uncertain. A high proportion of such labels, even if consistent, may indicate unreliability in the labeling process, particularly when the minority labels show lower agreement. Therefore, chance-adjusted correlation measures rely on key assumptions about the types of errors humans are likely to make. Furthermore, Krippendorff's-$\alpha$, specifies that the agreement coefficient is an indicator of reliability if and only if **(1)** the raters duplicate the process of rating, i.e., when two independent raters use the same guidelines or instructions and use the same units, **(2)** raters must be treated as interchangeable, and **(3)** correct answers are not known (Krippendorff, 2004). Human labels and machine labels are clearly not interchangeable even with current state-of-the-art LLMs to generate machine labels, as humans provide the ground truth labels.

### 2.2 DIFFERENCES BETWEEN CORRELATION MEASURES AND RAW ACCURACY

The assumptions above, especially the assumption that raters must be treated as interchangeable, raise a fundamental question as to why even use correlation measures to compare machine and human performance, instead of measures such as accuracy and F1 using human labels as ground truth? *Firstly*, one of the main challenges with using scores like accuracy or F1 is that human labels tend to be noisy regardless of the task or the dataset, where some label choices are more acceptable than others, and measures such as F1 or accuracy do not account for variations in human labels and hence correlation measures *appear to be a better measure*. However, as mentioned previously,

many of the common chance-adjusted correlation measures aim to account for **random chance** agreement in HH agreement, whereas the errors machines make tend to be more **systematic**. A key characteristic of random errors is that they are non-reproducible. When repeating the labeling process, the result varies, resulting in low IRA, which in-turn may point to potential problems in the labeling process or complexities in the task. In contrast, systematic errors are consistent, and IRA measures are not designed to detect them. *Secondly*, as a direct consequence, IRA metrics assume that agreement is good and variance is bad, which implies that there is a single correct answer – a situation that may not hold for ambiguous items or those that gauge user perception.

### 2.3 TYPES OF RAW DATA COLLECTED DURING EVALUATION

The type of the data collected, such as nominal, ordinal, interval, ratio, discrete or continuous data, dictate how such data can be aggregated and interpreted. The most common types of values collected during automatic or human evaluation are:

**Nominal values:** Nominal data, such as gender, have no natural ranking or ordering. Such data are primarily used in human evaluation tasks such as fact checking, where the aim is to verify whether a given text can be inferred from reference content (Liu et al., 2023c) and usually is modeled as a binary classification task. Preference judgments such as "*Is the output of model-A or Model-B better?*" are also common examples of nominal values collected during human evaluation. IRA measures for nominal values include percentage agreement (McHugh, 2012), Randolph's-$\kappa$, Fleiss's-$\kappa$, Krippendorff's-$\alpha$ and Cohen's-$\kappa$.

**Ordinal values:** Ordinal values have an implicit order, e.g., ratings of quality (very good, good, fair, poor, very poor) (Mishra et al., 2018). The most common use of ordinal values in human evaluation are through Likert scales. Likert scales are rating systems of attitude scale to measure the degree to which a person agrees/disagrees with a given statement (Taherdoost, 2019) and commonly used to gauge user perception on qualitative aspects such as readability, coherence, and fluency (van der Lee et al., 2021). IRA measures for ordinal values should ideally account for the fact that ratings that are closer to one another (1 vs. 2) for a given item indicate a higher agreement than ratings that are further apart (1 vs. 4). Such IRA measures include Krippendorff's-$\alpha$ for ordinal values, where disagreements are weighted by distance between the ratings. Rank correlations such as Kendall's-$\tau$ and Spearman's-$\rho$ are also widely used to measure item-level correlation between machines and humans for ordinal values. However, there's *a caveat on using these measures*: their ability to handle a relatively large proportion of ties is widely debated, even when measuring system correlations that involve a much smaller proportion of ties (Deutsch et al., 2023). When the number of items to rank is $n$, the number of ordinal values to assign is $r$, and when $r << n$, it results in many ties or rank collision. For instance, when assigning a Likert scale of 1-5 to 100 items, at least 20% of the items will have the same rank. While ordinal values preserve order, the objective of using ordinal values is not to rank the item in relation to other items, therefore the proportion of ties affects the interpretability and reliability of the rank correlation scores.

**Continuous values:** Continuous values are typically obtained when using automatic evaluation methods such as Rouge (Lin, 2004) and BERTScore (Zhang* et al., 2020).

## 3 ANALYSIS SETTINGS AND RESULTS

**Datasets and human labels:** We analyze typical evaluation tasks for LLM-generated text, including **(a)** Likert-based qualitative evaluation (ordinal), **(b)** NLI-style fact or consistency checking (nominal), and **(c)** human preferences of several LLMs (nominal). We select datasets with multiple human annotations: **1. SummEval** (Fabbri et al., 2021) is human evaluation of summary quality using four criteria – fluency, coherence, consistency, and relevance, with Likert scores assigned from 1 to 5. This dataset has 1600 samples, where each item is labeled 3 times by expert annotators. **2. SNLI** (Bowman et al., 2015) is an NLI inference task dataset with 10K samples and 3 labels – entailment, neutral and contradiction. Each item is labeled 5 times by human annotators. **3. MNLI** (Williams et al., 2018) is also an NLI inference task dataset with 10K samples and 3 labels – entailment, neutral and contradiction. Each item is assigned 5 labels by human annotators. It contains two splits – *matched* and *mismatched*. **4. MT-Bench** (Zheng et al., 2023) is a preference dataset which compares human preferences of several model outputs on open-ended questions using multi-turn conversations. From this dataset, we only use samples that have at least 3 human ratings. **5.** We also include **JudgeBench** (Bavaresco et al., 2024) in Appendix A.12.

**Machine labels:** For **SummEval**, we reuse the original G-Eval results (Liu et al., 2023b) which rates the quality of summaries on a scale of 1-5. In addition, we use Claude Sonnet 3 and Mistral

Large v2 on SummEval, where prompts are detailed in Appendix A.2. For **SNLI** and **MNLI**, we use Llama 3.1 8B, Mistral Large v2 and Sonnet 3, reusing the prompts from Liu et al. (2023a) as detailed in Appendix A.1. For **MT-Bench**, we rely on the existing GPT-4 labels in the dataset (Zheng et al., 2023). We also adapt the same prompts (see Appendix A.3) for Sonnet 3 as LJ. In order to compute MM correlation, we set the sampling temperature to 1.0 to maximize the variation across runs, and set top_p = 0.8. We sample 20 machine labels per example. **NOTE:** We have predominately reused the prompts from existing works of using GPT-4 as LJ, we have NOT attempted to optimize the prompts per LLM except for minor format changes. Since models are known to be highly sensitive to prompts, our experiments are not meant to rank the underlying LJ, but to study the impact of human uncertainty regardless of the LJ.

**Correlation measures:** For nominal data, we report scores on Krippendorff's-$\alpha$, Fleiss-$\kappa$, Randolph's-$\kappa$ and percentage agreement (McHugh, 2012) also detailed in Appendix A.4. For MT-Bench preference data, since the number of human raters vary per tuple $\langle$conversation, turn, model pair$\rangle$ we use Krippendorff-$\alpha$ and percentage agreement to compute HH correlation as it can handle varying number of raters per item unlike Fleiss-$\kappa$ or Randolph's-$\kappa$. For ordinal data, we report Krippendorff-$\alpha$ for ordinal data. We also report commonly used rank correlation measures such as Spearman's-$\rho$ and Kendall's-$\tau$, as many papers have used them to measure for item-level correlation (despite the caveat highlighted previously).

### 3.1    RQ1: How does uncertainty in human labels impact correlation metrics when we measure the efficacy of automatic evaluation methods?

To study the impact of human uncertainty, we stratify samples and then compare HH and Human-to-machine (HM) correlation measurements for each group. We primarily stratify *by percentage agreement*, which is the proportion of labels that matches the majority (for nominal data) or the median label (for ordinal data) among all human labels assigned to an item. For example, if 3 out of 5 people assign the same label, the percentage agreement is 60%. We also stratify samples *by the number of unique human labels* to ensure that our findings are consistent regardless of the stratification method.

In Table 1, column $H^w M^w$ compares Human majority ($H^w$) with machine majority ($M^w$). Superficially, HM correlation seems to improve with increasing human certainty. On closer inspection, another trend emerges where in any stratified group, when uncertain samples increases (as measured by low HH correlation), the HM correlation seems to outperform HH correlation as highlighted. However, as the proportion of samples with noisy labels decreases and HH correlation approaches near-perfect agreement, HH correlation is much higher than HM as highlighted. Thus, **the illusion that machines (specifically LJs) approximate majority human behavior is largely due to the presence of noisy labels in the data**. If LJs were truly approximating the human majority, the $\Delta$ would approach 0 under perfect HH correlation – which is clearly not the case. We also replicated this behavior on SummEval dataset which relies on ordinal labels in Table 2, using correlation metrics such as Krippendorff's-$\alpha$ for ordinal data, Kendall's-$\tau$ and Spearman's-$\rho$. The same patterns are observed on preference data in Table 3. We further randomly sample subsets from NLI datasets (sampling detailed in Appendix A.5) and compare correlation scores across subsets with varying levels of human label uncertainty. We repeat this for three different LLMs, and consistently observe the same findings, performance gaps become more apparent as human label consistency increases, as shown in Fig. 1. Similar findings for the SummEval dataset are shown in Appendix Fig. 5.

We explore the phenomenon using a random labeler on synthetically labeled datasets (code provided in Appendix A.11). This dataset simulates both binary classification and ordinal classification with labels $\{1, 2, 3\}$. We first sample 2 synthetic human labels to each item. Then another random labeler assigns labels randomly. We examine the correlation between the random labels and the synthetic human labels across varying levels of certainty. Notably, the random labeler may appear to have higher correlation when the proportion of uncertain samples is higher. However, as the proportion of consistently labeled samples increases, the random labels' poor performance becomes evident, as shown in Fig. 2. Intuitively, if 2 human labels disagree, a 3rd random label cannot perform worse in terms of agreement – they can either disagree with both human labels (in which case it is no better than the 2 humans) or agree with one of the human label. **This example further illustrates why stratification by proportion of uncertainty is crucial to show weaknesses in *any automatic labeler*.**

| Dataset | Partition by human labels (% samples) | Krippendorff's-$\alpha$ | | | | % Agreement | | | Fleiss-$\kappa$ | | | Randoph's-$\kappa$ | | |
|---|---|---|---|---|---|---|---|---|---|---|---|---|---|---|
| | | HH | MM | $H^wM^w$ | $\Delta$ | HH | $H^wM^w$ | $\Delta$ | HH | $H^wM^w$ | $\Delta$ | HH | $H^wM^w$ | $\Delta$ |
| MNLI (matched) | All (100%) | 0.70 | 0.96 | 0.72 | -0.02 | 0.88 | 0.82 | 0.07 | 0.70 | 0.72 | -0.02 | 0.70 | 0.72 | -0.03 |
| | PA = 1 (58%) | 1.00 | 0.97 | 0.83 | 0.17 | 1.00 | 0.89 | 0.11 | 1.00 | 0.83 | 0.17 | 1.00 | 0.83 | 0.17 |
| | $0.8 \le PA < 1$ (25%) | 0.39 | 0.95 | 0.66 | -0.27 | 0.80 | 0.78 | 0.02 | 0.39 | 0.66 | -0.27 | 0.40 | 0.67 | -0.27 |
| | $0 \le PA < 0.8$ (16%) | 0.04 | 0.93 | 0.37 | -0.33 | 0.60 | 0.61 | -0.01 | 0.04 | 0.37 | -0.33 | 0.07 | 0.42 | -0.34 |
| | unique = 2 (38%) | 0.28 | 0.95 | 0.56 | -0.28 | 0.73 | 0.72 | 0.01 | 0.28 | 0.56 | -0.28 | 0.30 | 0.58 | -0.28 |
| | unique = 3 (3%) | -0.06 | 0.91 | 0.39 | -0.45 | 0.60 | 0.63 | -0.03 | -0.06 | 0.39 | -0.45 | -0.05 | 0.44 | -0.49 |
| MNLI (mismatched) | All (100%) | 0.72 | 0.97 | 0.74 | -0.02 | 0.89 | 0.83 | 0.07 | 0.72 | 0.74 | -0.02 | 0.72 | 0.74 | -0.02 |
| | PA = 1 (60%) | 1.00 | 0.98 | 0.85 | 0.15 | 1.00 | 0.90 | 0.10 | 1.00 | 0.85 | 0.15 | 1.00 | 0.85 | 0.15 |
| | $0.8 \le PA < 1$ (24%) | 0.39 | 0.95 | 0.65 | -0.26 | 0.80 | 0.77 | 0.03 | 0.39 | 0.65 | -0.26 | 0.40 | 0.66 | -0.26 |
| | $0 \le PA < 0.8$ (14%) | 0.04 | 0.94 | 0.39 | -0.35 | 0.60 | 0.62 | -0.02 | 0.04 | 0.39 | -0.35 | 0.07 | 0.43 | -0.36 |
| | unique = 2 (36%) | 0.28 | 0.94 | 0.56 | -0.28 | 0.73 | 0.72 | 0.01 | 0.28 | 0.56 | -0.28 | 0.30 | 0.58 | -0.28 |
| | unique = 3 (2%) | -0.06 | 0.94 | 0.41 | -0.46 | 0.60 | 0.64 | -0.04 | -0.06 | 0.41 | -0.46 | -0.05 | 0.45 | -0.50 |
| SNLI | All (100%) | 0.68 | 0.97 | 0.63 | 0.05 | 0.88 | 0.76 | 0.12 | 0.68 | 0.63 | 0.05 | 0.68 | 0.64 | 0.05 |
| | PA = 1 (55%) | 1.00 | 0.97 | 0.69 | 0.31 | 1.00 | 0.80 | 0.20 | 1.00 | 0.69 | 0.31 | 1.00 | 0.69 | 0.31 |
| | $0.8 \le PA < 1$ (28%) | 0.39 | 0.97 | 0.62 | -0.24 | 0.80 | 0.76 | 0.04 | 0.39 | 0.62 | -0.23 | 0.40 | 0.64 | -0.24 |
| | $0 \le PA < 0.8$ (15%) | 0.03 | 0.95 | 0.37 | -0.34 | 0.60 | 0.61 | -0.01 | 0.03 | 0.37 | -0.34 | 0.07 | 0.42 | -0.35 |
| | unique = 2 (41%) | 0.29 | 0.96 | 0.54 | -0.25 | 0.74 | 0.71 | 0.03 | 0.29 | 0.54 | -0.25 | 0.31 | 0.57 | -0.26 |
| | unique = 3 (3%) | -0.05 | 0.94 | 0.47 | -0.53 | 0.60 | 0.66 | -0.06 | -0.05 | 0.47 | -0.53 | -0.05 | 0.50 | -0.55 |

Table 1: Compare HM and HH correlation measurements on NLI datasets using Sonnet to generate machine labels. The overall correlation scores (All) indicate that the HM correlation is similar to HH. However, when stratifying by percentage agreement (PA) or human unique labels, the difference ($\Delta = HH - H^wM^w$) is highest when the all human labels are the same (PA=1), and $H^wM^w$ outperforms HH when certainty is the lowest. $H^w$ and $M^w$ are human and machine majority label.

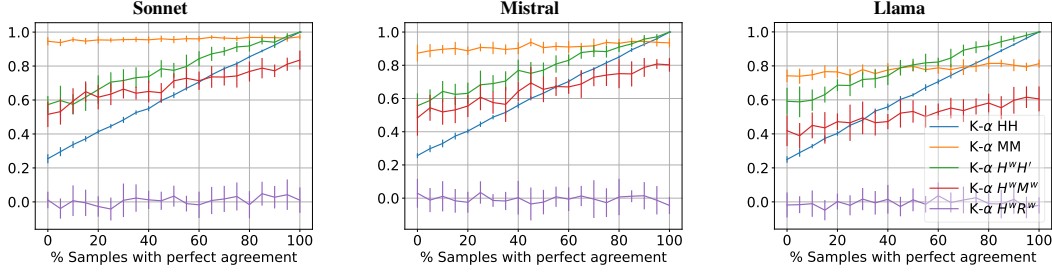

Figure 1: Krippendorff's $\alpha$ (K-$\alpha$) on subsets of MNLI-mismatched with varying uncertainty, using 3 LLMs – Sonnet, Mistral, and Llama. As proportion of samples with consistent human labels increases (towards the right of X-axis), HH correlations are significantly higher than $H^wM^w$. $H^wH'$ represents the correlation between the human majority and another human annotator ($H'$). $H^wR^w$ compares the human majority with a random machine evaluator, serving as baseline performance.

## 3.2 RQ2: HOW CAN WE MEASURE HUMAN-MACHINE AGREEMENT THAT ACCOUNTS FOR HUMAN UNCERTAINTY AS A RESULT OF VARIATION IN HUMAN PERCEPTION?

Evaluating the quality of a free-form model generated output, such as a summary, largely depends on human perception. As a result, the human ratings can vary depending on a wide range of factors including what one likes to their emotional state (Elangovan et al., 2024). Perception-based ordinal-value evaluation schemes, such as Likert scores, pose 2 main challenges. **(1)** It is an attitude-based scale, and since human attitudes vary, metrics like Krippendorff's $\alpha$, which penalize variation, can produce scores lower than the minimum acceptable value (Amidei et al., 2019). **(2)** Rank correlation measures such as Kendall-$\tau$ or Spearman's-$\rho$ are not entirely appropriate given the large number of ties when comparing median scores, as previously discussed in Sec. 2.3 and in the works by Deutsch et al. (2023). Furthermore, whether to treat Likert-data as ordinal or interval data dictates the aggregation method – median or mean, is also debated (Joshi et al., 2015). The argument against using Likert-data as interval data is that the points on the scale may not be equidistant, e.g., the distance between pair ⟨neutral, agree⟩ may not be the same as the distance between ⟨agree, strongly agree⟩. We report Spearman's-$\rho$ for both to illustrate the difference in Table 2. While in most cases the two sets of scores look largely the same, there are instances where the numbers are vastly different.

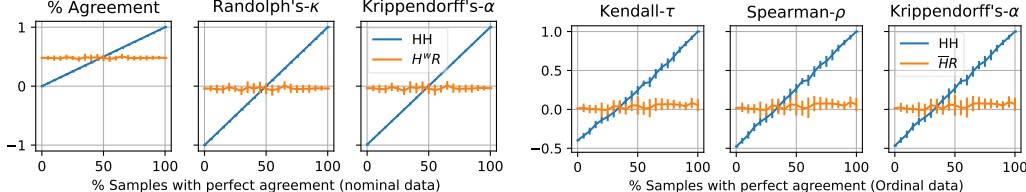

Figure 2: Simulating the impact of uncertainty by comparing with an automatic random labeler (**R**). When the proportion of samples with uncertainty is higher, even a random labeler can *appear* to have better correlation with a majority ($H^w$) or median ($\overline{H}$) human label. As the proportion of samples with consistency increases, the weakness of the random labeler become evident.

| C | Partition (% samples) | Rouge1 ρ | Rouge1 τ | Model | Krippendorff's-α HH | MM | $\overline{HM}$ | Δ | $H'\overline{M}$ | $JS_b$ HM | Spearman's-ρ $\overline{HM}$ | $H'\overline{M}$ | $H^\mu M^\mu$ | Kendall's-τ $\overline{HM}$ | $H'\overline{M}$ |
|---|---|---|---|---|---|---|---|---|---|---|---|---|---|---|---|
| **Coh** | 1. All (100%) | 0.18 | 0.14 | G-Eval | 0.55 | 0.74 | 0.31 | 0.25 | 0.27 | 0.39 | 0.53 | 0.43 | 0.57 | 0.47 | 0.38 |
| | | | | Mistral | 0.55 | 0.78 | 0.49 | 0.07 | 0.42 | 0.36 | 0.56 | 0.48 | 0.62 | 0.49 | 0.42 |
| | | | | Sonnet | 0.55 | 0.83 | 0.24 | 0.32 | 0.21 | 0.51 | 0.34 | 0.31 | 0.41 | 0.30 | 0.28 |
| | 2. PA = 1 (12%) | 0.36 | 0.28 | G-Eval | 1.00 | 0.78 | 0.25 | 0.75 | 0.25 | 0.70 | 0.61 | 0.61 | 0.64 | 0.55 | 0.55 |
| | | | | Mistral | 1.00 | 0.79 | 0.35 | 0.65 | 0.35 | 0.69 | 0.62 | 0.62 | 0.68 | 0.56 | 0.56 |
| | | | | Sonnet | 1.00 | 0.86 | 0.24 | 0.76 | 0.24 | 0.71 | 0.43 | 0.43 | 0.48 | 0.39 | 0.39 |
| | 3. PA < 60% (21%) | 0.17 | 0.13 | G-Eval | 0.19 | 0.70 | 0.32 | -0.13 | 0.16 | 0.35 | 0.56 | 0.31 | 0.58 | 0.52 | 0.27 |
| | | | | Mistral | 0.19 | 0.78 | 0.55 | -0.37 | 0.36 | 0.30 | 0.57 | 0.39 | 0.64 | 0.52 | 0.34 |
| | | | | Sonnet | 0.19 | 0.85 | 0.17 | 0.02 | 0.14 | 0.50 | 0.40 | 0.31 | 0.45 | 0.37 | 0.28 |
| **Rel** | 1. All (100%) | 0.33 | 0.26 | G-Eval | 0.40 | 0.74 | 0.25 | 0.15 | 0.22 | 0.37 | 0.53 | 0.42 | 0.59 | 0.47 | 0.37 |
| | | | | Mistral | 0.40 | 0.79 | 0.38 | 0.02 | 0.30 | 0.31 | 0.54 | 0.42 | 0.61 | 0.49 | 0.38 |
| | | | | Sonnet | 0.40 | 0.79 | 0.35 | 0.05 | 0.29 | 0.38 | 0.39 | 0.34 | 0.48 | 0.36 | 0.31 |
| | 2. PA = 1 (14%) | 0.38 | 0.30 | G-Eval | 1.00 | 0.76 | 0.30 | 0.70 | 0.30 | 0.63 | 0.69 | 0.69 | 0.73 | 0.63 | 0.63 |
| | | | | Mistral | 1.00 | 0.80 | 0.43 | 0.57 | 0.43 | 0.59 | 0.71 | 0.71 | 0.72 | 0.65 | 0.65 |
| | | | | Sonnet | 1.00 | 0.77 | 0.36 | 0.64 | 0.36 | 0.51 | 0.49 | 0.49 | 0.60 | 0.45 | 0.45 |
| | 3. PA < 60% (19%) | 0.28 | 0.22 | G-Eval | -0.09 | 0.69 | 0.18 | -0.27 | 0.09 | 0.33 | 0.40 | 0.17 | 0.42 | 0.38 | 0.15 |
| | | | | Mistral | -0.09 | 0.74 | 0.32 | -0.41 | 0.16 | 0.28 | 0.42 | 0.19 | 0.46 | 0.40 | 0.17 |
| | | | | Sonnet | -0.09 | 0.75 | 0.29 | -0.38 | 0.16 | 0.43 | 0.36 | 0.22 | 0.40 | 0.35 | 0.20 |

Table 2: Results on SummEval dataset for criteria – **Coh**erence and **Rel**evance, $\overline{H}$ is the human median rating and $\overline{M}$ is machine median. $\Delta = (HH - \overline{HM})$. Impact of human uncertainty: When the human label uncertainty is high, $\overline{HM}$ can seem better than HH correlation and vice versa under low uncertainty. Does the correlation score ($\rho$ of 0.56 , $\alpha$ 0.49) imply the LJ is good enough to replace human evaluation? Spearman's-$\rho$ median ($\overline{HM}$) vs. mean ($H^\mu M^\mu$): Median vs. mean results in an 11 points boost $0.49 \rightarrow 0.60$. Differences in correlation metrics: Krippendorff's-$\alpha$ of 0.25 (reject results), while Spearman's-$\rho$ shows 0.53 (moderate correlation). LJ ranking and absolute correlation numbers change depending on reference human under high uncertainty, Mistral best performance 0.42 replaced by Sonnet 0.22 when reference human is $H'$ (another human annotator). Full table in Appendix Table 7.

For instance, Table 2 Spearman's-$\rho$ median vs. mean, median comparison yields a correlation of 0.49 (low correlation), while comparing the mean results in 0.60 (indicating moderate correlation) (Mukaka, 2012), provoking the question which number is more representative of the underlying data? Regardless of whether we choose to use Krippendorff-$\alpha$ or a rank correlation metric, one aspect that both assume is that there is a single rating (e.g., median value) that appropriately represents the underlying choice a human rater makes. This assumption may not be correct, especially when the task is inherently subjective, especially when the number of judgments collected per item is relatively low, e.g., 3 ratings per item. This shows a clear gap in existing metrics to compare human and machine performance, where intra-human ratings naturally vary as a a result of subjectivity inherent to the task.

Hence, we propose an additional metric to supplement existing measures, the use of binned Jensen-Shannon divergence (JSD) for perception (Lin, 1991) to quantify how closely automatic methods mimic human judgment. To the best of our knowledge, our approach of using JSD has not been used as a correlation measure to account for human perception, without relying on a single gold label. The closest work to ours is the use of cumulative JSD to predict the aesthetic score distribution, instead of just the average rating, to represent image quality (Jin et al., 2018).

| MP (A vs B) | LJ | Partition (# samples) | Human | Krippendorff's-$\alpha$ HH | MM | $H^wM^w$ | $\Delta$ | $H'M^w$ | % Agreement HH | $H^wM^w$ | $H'M^w$ | $JS_b$ HM | Fleiss-$\kappa$ $H^wM^w$ | $H'M^w$ | Randolph-$\kappa$ $H^wM^w$ | $H'M^w$ | LJ Winrate W | WR | $\Delta$ |
|---|---|---|---|---|---|---|---|---|---|---|---|---|---|---|---|---|---|---|---|
| ⟨Alp, Gp3⟩ | GPT | 1. All (52) | B 0.79 | 0.24 | - | 0.32 | -0.08 | 0.29 | 0.81 | 0.77 | 0.75 | 0.11 | 0.31 | 0.28 | 0.65 | 0.62 | B | 0.81 | 0.02 |
| | | 2. PA = 1 (26) | B 0.92 | 1.00 | - | -0.05 | 1.05 | -0.05 | 1.00 | 0.85 | 0.85 | 0.13 | -0.07 | -0.07 | 0.77 | 0.77 | B | 0.92 | 0.00 |
| | | 3. PA: [60,80) (18) | B 0.61 | -0.04 | - | 0.36 | -0.40 | 0.28 | 0.68 | 0.67 | 0.61 | 0.22 | 0.34 | 0.26 | 0.50 | 0.42 | B | 0.67 | 0.06 |
| | Sonnet | 1. All (52) | B 0.79 | 0.24 | 0.90 | 0.01 | 0.23 | 0.11 | 0.81 | 0.73 | 0.73 | 0.18 | 0.00 | 0.10 | 0.60 | 0.60 | B | 0.90 | 0.12 |
| | | 2. PA = 1 (26) | B 0.92 | 1.00 | 1.00 | -0.02 | 1.02 | -0.02 | 1.00 | 0.92 | 0.92 | 0.17 | -0.04 | -0.04 | 0.85 | 0.85 | B | 1.00 | 0.08 |
| | | 3. PA: [60,80) (18) | B 0.61 | -0.04 | 0.93 | -0.04 | 0.00 | 0.18 | 0.68 | 0.50 | 0.61 | 0.17 | -0.07 | 0.16 | 0.25 | 0.42 | B | 0.78 | 0.17 |
| ⟨Cld, Gp3⟩ | GPT | 1. All (36) | A 0.39 | 0.40 | - | 0.54 | -0.14 | 0.34 | 0.79 | 0.69 | 0.56 | 0.11 | 0.54 | 0.33 | 0.54 | 0.33 | A | 0.39 | 0.00 |
| | | 2. PA = 1 (14) | B 0.64 | 1.00 | - | 0.25 | 0.75 | 0.25 | 1.00 | 0.57 | 0.57 | 0.10 | 0.22 | 0.22 | 0.36 | 0.36 | B | 0.57 | 0.07 |
| | | 3. PA: [60,80) (21) | A 0.52 | -0.02 | - | 0.58 | -0.60 | 0.35 | 0.67 | 0.76 | 0.57 | 0.12 | 0.57 | 0.34 | 0.64 | 0.36 | A | 0.52 | 0.00 |
| | Sonnet | 1. All (36) | A 0.39 | 0.40 | 0.78 | 0.36 | 0.05 | 0.18 | 0.79 | 0.58 | 0.47 | 0.17 | 0.35 | 0.17 | 0.38 | 0.21 | A | 0.56 | 0.17 |
| | | 2. PA = 1 (14) | B 0.64 | 1.00 | 0.56 | -0.07 | 1.07 | -0.07 | 1.00 | 0.43 | 0.43 | 0.24 | -0.11 | -0.11 | 0.14 | 0.14 | B | 0.64 | 0.00 |
| | | 3. PA: [60,80) (21) | A 0.52 | -0.02 | 0.86 | 0.37 | -0.39 | 0.09 | 0.67 | 0.67 | 0.48 | 0.19 | 0.35 | 0.07 | 0.50 | 0.21 | A | 0.67 | 0.14 |
| ⟨Gp3, Gp4⟩ | GPT | 1. All (38) | B 0.53 | 0.26 | - | 0.27 | -0.01 | 0.22 | 0.74 | 0.61 | 0.61 | 0.18 | 0.26 | 0.21 | 0.41 | 0.41 | B | 0.74 | 0.21 |
| | | 2. PA = 1 (11) | B 0.82 | 1.00 | - | 0.64 | 0.36 | 0.64 | 1.00 | 0.91 | 0.91 | 0.16 | 0.63 | 0.63 | 0.86 | 0.86 | B | 0.91 | 0.09 |
| | | 3. PA: [60,80) (20) | B 0.50 | 0.08 | - | 0.24 | -0.17 | 0.12 | 0.68 | 0.55 | 0.50 | 0.21 | 0.22 | 0.10 | 0.33 | 0.25 | B | 0.65 | 0.15 |
| | Sonnet | 1. All (38) | B 0.53 | 0.26 | 0.76 | 0.20 | 0.06 | 0.15 | 0.74 | 0.50 | 0.50 | 0.13 | 0.19 | 0.14 | 0.25 | 0.25 | B | 0.45 | 0.08 |
| | | 2. PA = 1 (11) | B 0.82 | 1.00 | 0.66 | 0.19 | 0.81 | 0.19 | 1.00 | 0.64 | 0.64 | 0.28 | 0.15 | 0.15 | 0.45 | 0.45 | B | 0.64 | 0.18 |
| | | 3. PA: [60,80) (20) | B 0.50 | 0.08 | 0.81 | 0.22 | -0.15 | 0.19 | 0.68 | 0.50 | 0.50 | 0.06 | 0.20 | 0.17 | 0.25 | 0.25 | B | 0.45 | 0.05 |

Table 3: Stratified performance on preference data with at least 3 human ratings from MT-Bench for model pairs (**MP**). Impact of human label uncertainty: When the HH label uncertainty is high, $H^wM^w$ can seem better than HH correlation and vice versa. IRA vs. Win-rate: While the difference in win-rate between model and human ($\Delta$) can be smaller in one evaluator model (e.g., Sonnet 0.08) compared to another (e.g., GPT 0.21), the $H^wM^w$ IRA can tell an opposite story, as in the example above, GPT has better correlation with human majority across all correlation metrics except $JS_b$. LJ ranking and absolute correlation numbers change depending on reference human H', mimicking single annotator, under high uncertainty compared to using consensus human majority ($H^w$).

Formally (source code in Appendix A.7), let $H_i$ and $M_i$ represent the set of human and machine ratings respectively assigned to the item $x_i$, and $n$ be the total number of the items and $R$ be the set of possible ratings (e.g., $1$ to $5$ for ordinal ratings, or $\{A, B, Tie\}$ for nominal model preferences). To compute the distance between human and machine predictions, we can define the probability mass function (PMF) for a given rating $c$ as follows:

$$P_r^h(c) = \frac{\sum_{i \in B_r} \sum_{l \in H_i} \mathbb{1}_{\{l=c\}}}{\sum_{i \in B_r} |H_i|} \text{ (Human)}, \qquad P_r^m(c) = \frac{\sum_{i \in B_r} \sum_{l \in M_i} \mathbb{1}_{\{l=c\}}}{\sum_{i \in B_r} |M_i|} \text{ (Machine)} \qquad (1)$$

where for ordinal values, bin $\mathbf{B}_r := \{i \in \{1,..n\} | \text{median}(H_i) = r\}$ contains the set of samples with a median human rating $r$. For nominal perception-based preference labels, bin $\mathbf{B}_r := \{i \in \{1,..n\} | \text{mostfrequent}(H_i) = r\}$ contains the set of samples with the most frequent human rating $r$.

From Equation 1, we can define the binned JSD ($JS_b$) as follows:

$$JS_b = \sum_{r \in R} \frac{|B_r|}{n} JS(P_r^h || P_r^m) := \sum_{r \in R} \frac{|B_r|}{n} \left( \frac{1}{2} KL(P_r^h || Q_r) + \frac{1}{2} KL(P_r^m || Q_r) \right), \qquad (2)$$

where $KL(P_r^h || Q_r) = \sum_{c \in R} P_r^h(c) \log_2(\frac{P_r^h(c)}{Q_r(c)})$ is the Kullback-Leibler divergence (Kullback & Leibler, 1951) and $Q_r(c) = \left( P_r^h(c) + P_r^m(c) \right)/2$ for $\forall c \in R$ is the average between human and machine PMFs, based on the definition of Jensen-Shannon distance ($JS$). Since JSD is a distance-based measure, lower scores are better because they indicate that the human and machine judgments are similar. We also illustrate the benefits of binned-JSD using a toy example in Appendix A.8.

The binned JSD for ordinal perception data has 2 main advantages, **(1)** it measures how closely machine labels mimic human perception without the need for a single gold label **(2)** human and machine labels are *not* treated interchangeably, as the items in a given bin are selected by the human median or majority value. For HM correlation, the proposed approach can be applied at item-level when the number of labels collected per item is high enough to form a distribution, however, for judging model generated output even obtaining 3 human labels per item can be expensive. Hence, we suggest computing the distribution distance per bin. The main shortcoming of this approach is that some key item level discrepancies between machines and humans ratings might not be clearly surfaced. For instance, for a given item, if all the humans assign a rating of 1 and machines assign say 4, at an aggregate level this pattern may not be obvious if this pattern is not frequent. Hence, we also recommend reporting correlations stratified by uncertainty in human labels, e.g., for items that have high agreement labels, such as perfect human agreement as shown in Table 2.

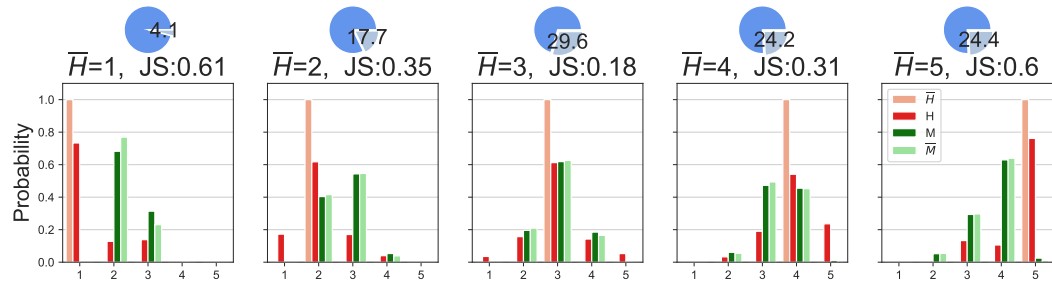

Figure 3: **Ordinal HM perception comparison chart:** Visualization of human perception vs. machine labels binned by human median rating (LJ Mistral on Coherence): Dial at the top shows the percentage of samples that fall into the bin. Middle scores (median 2-4) have higher human uncertainty, where only 60% (less than 2 out of 3) of the human labels follow the median. The extremes scores (1, 5) have less uncertainty. While Mistral seems to mimic human perception when the human median is 3, it is also biased towards the rating 3 when the human ratings are between 2-4.

### 3.3 RQ3: How can we visualize the underlying data to draw meaningful insights when we compare automatic labels and human labels?

#### 3.3.1 Perception-based ordinal rating visualization

We propose the "**Ordinal HM perception comparison chart**" as shown in Fig. 3 to compare humans' perception of the content vs. how machines rate them. Here, we bin the samples by the human median rating and plot the corresponding distribution of human and machine ratings. For instance, say for a grading task using label values between 1-5, each item is rated 3 times by humans. For a sample set of 10 items, the total number of judgments would be 30 (10 items * 3 ratings per item). Let's say 7 items get a median rating of 4, these 7 items would fall in bin $\overline{H} = 4$. We then plot the human label distribution for that bin to include all the 21 judgments (7 items * 3 ratings per item = 21), we also plot all the machine labels for the corresponding bin. We repeat this for each of the label from 1-5 as shown in Fig. 3. This allows us to visualize the variation in human perception, thus  including all the ratings rather than a single gold rating, and compare how the corresponding machine labels vary.

Correlation numbers are challenging to interpret, and do not provide sufficient insights to the meaning behind the numbers unless supported by visualization. When does a correlation score imply the model is good enough to replace human evaluation as indicated in Table 2? And what are the gaps in the LJ? In Table 2, LJ Mistral achieves a Krippendorff's-$\alpha$ $\overline{HM}$ of 0.49. Visualization in Fig. 3 shows that the LJ does not rate any item as 1 and negligible amount of items are rated 5, while humans have assigned over 24% of the items a median rating of 5. This shows the key difference between LJ Mistral's judgments and humans. This type of insight can potentially be useful in optimizing the prompts used by the LJ to minimize the gap with humans judgments, demonstrating the importance of effective visualization techniques.

Another key aspect to note is how human uncertainty varies across different median labels. Human labels tend to be more certain when they assign extreme ratings (1, 5) compared to the middle ratings (2-4), demonstrated by the distribution difference between the median $\overline{H}$ and all $H$ ratings, as shown in Fig. 3. Relatively higher consistency in extreme ratings is a common pattern observed when humans are asked to rate using star-rating schemes and Likert scales, an observation typically reported in recommendation systems (Amatriain et al., 2009). This type of differences in the extent of human uncertainty depending on the rating further demonstrates the deficiencies in assuming a single gold rating for tasks that rely on human perception, clearly illustrating the need for a HM correlation metric that does not assume a single gold label, such as the $JS_b$ we proposed in Sec. 3.2 in RQ2.

#### 3.3.2 Perception-based preference visualization

For visualizing pairwise model preferences to compare human and machine judgments, we propose the "**Pairwise preference HM perception chart**". The concept behind this visualization is similar to the Ordinal HM perception comparison chart, the main difference is that is bins are separated by

the most frequent human label for a given item. This visualization also helps understand, for a given model pair, how reliable the human majority is. For instance, in Fig. 4, in the case of ⟨claude-v1, gpt-3.5-turbo⟩, even though the preferred model is Claude ( 38.9% preference), when human majority prefers GPT-3.5 there is almost no contention. We also visualize the preference of LJ Claude Sonnet shown in Appendix Fig. 7.

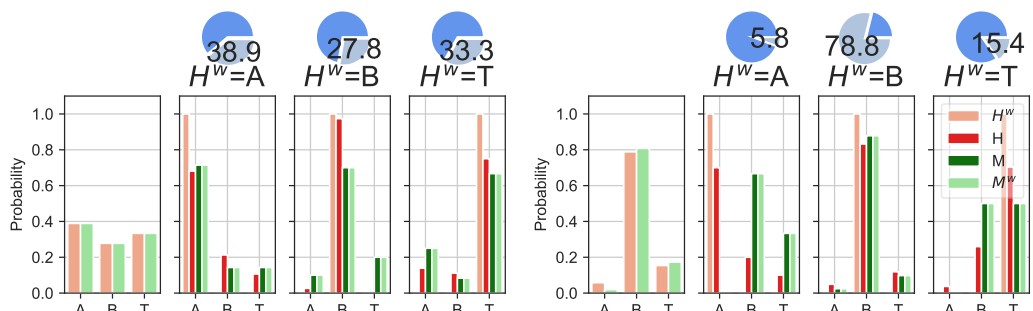

Figure 4: **Pairwise preference HM perception chart:** Human preference distribution vs. LJ GPT-4 for a given majority label per model pair. **T** is tie. Compares model pair *(Left)* Claude-v1 (**A**) vs. GPT3.5-turbo (**B**) *(Right)* Alpaca-13B (**A**) vs. GPT-3.5-turbo (**B**). Note: Using GPT-4 LJ has only one judgment per item in the results from Zheng et al. (2023), hence M$^w$ and M are the same.

## 4 DISCUSSION AND CONCLUSION

### 4.1 HUMANS ARE UNCERTAIN AND THEIR LABELS ARE NOISY, SO MACHINES ARE BETTER?!

Evaluation procedures have predominantly circumvented dealing with human uncertainty, except for recent limited works (Chen et al., 2020; Wang et al., 2023; Chen et al., 2024), and have thus relied on the simplified assumption that a single gold label is sufficient. A single gold label is only sufficient when there is little or no ambiguity, while being able to quantify human uncertainty is crucial in measuring the effectiveness of automatic evaluation. The ubiquitous nature of uncertainty is studied in human psychology (Kahneman et al., 2021) and is also illustrated in a widely used MNLI dataset, which was carefully curated with 5 human labels per item. Over 40% of the samples have some degree of uncertainty, as shown in Table 1. In addition, given the challenges in human evaluation, including obtaining consistent labels (Elangovan et al., 2024), recent datasets have started to rely on collecting one human label per item (Bai et al., 2022; Ganguli et al., 2022; Stiennon et al., 2020). This further demonstrates the urgent need to acknowledge that quantifying human uncertainty is essential to measure the effectiveness of automated evaluation.

The machine learning community cannot afford to dismiss uncertainty as simply *"poor quality labels from humans"*, key systematic errors and performance gaps in models become apparent under high human certainty as shown in Sec. 3.1. As a corollary, under high uncertainty even a random labeler can appear to have similar or better correlation with humans, as shown in Fig. 2. More importantly, for safety critical applications such as medicine, when humans labels are highly robust and certain, but the corresponding machine assigned labels differ, it can point to serious deficiencies in the automated system. Furthermore, the implications of uncertainty include changes to ranking of the automatic evaluator as well as the corresponding absolute correlation scores as shown in Table 2 and Table 3 depending on the reference human label. We acknowledge that collecting high quality labels from humans is expensive, and that in some cases gathering just one label per item might be adequate, such as when attempting to understand if one LLM is better than the other, provided the humans end up overwhelmingly preferring the output of one LLM over the other. In the case where the performance of two LLMs are close to one another, uncertainty in human judgments may be a signal that says the model outputs are quite similar. Thus, in some cases uncertainty can be a potential indicator of lower quality of labels requiring improvements to the underlying label collection process (Elangovan et al., 2024), in other cases uncertainty might be inevitable and therefore a valuable signal. Under uncertainty, comparison of automatic evaluation to human evaluation is challenging, and traditional correlation measures also fail to adequately account for this. The proposed binned JSD for perception is an effort in that direction so that metrics do not penalize human uncertainty, whilst comparing machine perception or attitudes with humans.

### 4.2 CHALLENGES IN METRICS AND INTERPRETABILITY

Deciding which statistic is appropriate depends on the data and how well it fits the assumptions made by the statistical measure (Eubanks, 2017). IRA metrics were never designed for systematic errors, and any accommodation for errors were based on assumptions about typical human behavior. Errors made by LLMs are rarely predictable, yet they are not random; rather, they are reproducible, making them systematic errors. The unpredictable nature of LLMs makes it difficult to design an effective metric that compares them with humans, given the uncertainty associated with human labels.

Despite the deficiencies in metrics, some metrics may be a better fit, depending on the aim and the results of the study. For instance, in the case of perception-based preference experiments, where the goal is to understand which among a given pair of models is better, and when the result is almost unanimous with one model overwhelmingly preferred over the other, then the agreement in rarer (minority) labels may not matter. Hence, measures such as Krippendorff's-$\alpha$ and Fleiss-$\kappa$ which estimate chance agreement based on the observed label distribution can result in substantially lower IRA scores even when there is high agreement on the majority label (winning model), whereas Randolph's-$\kappa$ assumes that the labels are equally distributed and hence is a better fit. For perception-based ordinal values, the extreme labels (such as 1 or 5 in the case of Likert 1-5) are usually strong indicators of human preferences, especially when supported by visualization as shown in Fig. 3. These extreme labels, might form the minority case, but might be crucial to interpreting the results. Hence, measures such as Krippendorff's-$\alpha$ and Fleiss-$\kappa$ estimate chance agreement based on the observed label distribution are a better fit compared to measures such as Randolph's-$\kappa$ that assumes uniform distribution of all labels. For tasks where a single majority label is not sufficient to represent human preferences, measures that do not assume a single gold label such as the proposed approach – binned JSD for perception, can be a better choice to compare human and machine performance.

In addition, statistical analysis, such as null-hypothesis and significance testing, is essential for determining whether one model outperforms another by random chance. Here, the chance component includes 2 obvious aspects, **(1)** chance due to the nature of samples in the evaluation set **(2)** uncertainty in human labels. A third aspect, even harder, is estimating the error rate as a result of systematically unpredictable erroneous labels from any automated evaluator. Future studies should explore these problems, including approaches like resampling (Deutsch et al., 2021). Incorporation of chance in rank correlation is also an important aspect to account for when two models differ in rank, but the corresponding difference in absolute scores is negligible, then the difference in the rank may not be meaningful.

IRA metrics are challenging to interpret, and hence visualization is key to understanding gaps and strengths of any given metric. Effective visualization is a trade-off between plotting every single data point (too much information that is hard to synthesize) and an aggregate view (summarized view where key information might be obscured). The proposed perception charts are a step towards emphasizing how an aggregate number may not be sufficient in capturing the underlying data, depicting how human uncertainty varies across different labels, which in turn affects how machine performance can be interpreted across labels as shown in Fig. 3 and 4.

### 4.3 RECOMMENDATIONS FOR REPORTING EFFECTIVENESS OF AUTOMATIC METHODS

To conclude, comprehensive analysis of performance involves investigation beyond a single aggregate number, also demonstrated in an additional case study in Appendix A.12. To that effect, in the case of comparing automatic evaluation with human evaluation, we recommend the following steps:
**1. Stratification by uncertainty levels:** As discussed in Sec. 3.1, uncertainty in human labels can obfuscate performance gaps between machines and human evaluators. Hence, we strongly recommend stratifying results by uncertainty proportions.
**2. Multi-metric reporting:** If there was no uncertainty, measures such as F1 would have worked. However, as a result of uncertainty, no single metric can capture important insights about every type of data as demonstrated in Sections 3.1, 3.2 and 3.3. Thus, we recommend reporting on multiple metrics belonging to different families, such as chance and non-chance-adjusted measures, so each metric in its own way can assist in bringing the less obvious but critical aspects about the underlying data to the forefront.
**3. Visualization of results:** A single non-parametric aggregate metric can rarely capture the entirety of underlying raw data, and hence visualization is key to understanding performance gaps, as discussed in Section 3.3. The proposed perception charts are a step towards making aggregate correlation more interpretable, as well as highlighting the strengths and gaps of the automatic labeler.

ACKNOWLEDGMENTS

We would like to thank Saab Mansour for reviewing our paper and providing valuable feedback.

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

# A  APPENDIX

## A.1  PROMPTS USED FOR NLI

These prompts are reused from Liu et al. (2023a)

```
You will be presented with a premise and a hypothesis about that premise. You need to decide
whether the hypothesis is entailed by the premise by choosing one of the following answers:

[[e]]: The hypothesis follows logically from the information contained in the premise.
[[n]]: It is not possible to determine whether the hypothesis is true or false without further
 information.
[[c]]: The hypothesis is logically false from the information contained in the premise.

Read the following premise and hypothesis thoroughly and select the correct answer from the
three answer labels.

Premise:
{premise}

Hypothesis:
{hypothesis}

Make a selection from "[[e]]", "[[n]]", "[[c]]". Only write the answer, do not write reasons.
```

## A.2  PROMPTS USED FOR SUMMEVAL

Note: These prompts are reused from Liu et al. (2023b) used in G-Eval.

### COHERENCE

```
You will be given one summary written for a news article.

Your task is to rate the summary on one metric.

Please make sure you read and understand these instructions carefully. Please keep this
document open while reviewing, and refer to it as needed.

Evaluation Criteria:

Coherence (1-5) - the collective quality of all sentences. We align this dimension with the
DUC quality question of structure and coherence whereby "the summary should be well-structured
 and well-organized. The summary should not just be a heap of related information, but should
build from sentence to a coherent body of information about a topic."

Evaluation Steps:

1. Read the news article carefully and identify the main topic and key points.
2. Read the summary and compare it to the news article. Check if the summary covers the main
topic and key points of the news article, and if it presents them in a clear and logical order
.
3. Assign a score for coherence on a scale of 1 to 5, where 1 is the lowest and 5 is the
highest based on the Evaluation Criteria.

Example:

Source Text:

{document}

Summary:

{summary}

Evaluation Form output the score only:
```

### CONSISTENCY

```
You will be given a news article. You will then be given one summary written for this article.

Your task is to rate the summary on one metric.
```

```
Please make sure you read and understand these instructions carefully. Please keep this
document open while reviewing, and refer to it as needed.

Evaluation Criteria:

Consistency (1-5) - the factual alignment between the summary and the summarized source. A
factually consistent summary contains only statements that are entailed by the source document
. Annotators were also asked to penalize summaries that contained hallucinated facts.

Evaluation Steps:

1. Read the news article carefully and identify the main facts and details it presents.
2. Read the summary and compare it to the article. Check if the summary contains any factual
errors that are not supported by the article.
3. Assign a score for consistency based on the Evaluation Criteria.

Example:

Source Text:

{document}

Summary:

{summary}

Evaluation Form output the score only:
```

## FLUENCY

```
You will be given one summary written for a news article.

Your task is to rate the summary on one metric.

Please make sure you read and understand these instructions carefully. Please keep this
document open while reviewing, and refer to it as needed.

Evaluation Criteria:

Fluency (1-3): the quality of the summary in terms of grammar, spelling, punctuation, word
choice, and sentence structure.

1: Poor. The summary has many errors that make it hard to understand or sound unnatural.
2: Fair. The summary has some errors that affect the clarity or smoothness of the text, but
the main points are still comprehensible.
3: Good. The summary has few or no errors and is easy to read and follow.

Example:

Summary:

{summary}

Evaluation Form output the score only:
```

## RELEVANCE

```
You will be given one summary written for a news article.

Your task is to rate the summary on one metric.

Please make sure you read and understand these instructions carefully. Please keep this
document open while reviewing, and refer to it as needed.

Evaluation Criteria:

Relevance (1-5) - selection of important content from the source. The summary should include
only important information from the source document. Annotators were instructed to penalize
summaries which contained redundancies and excess information.
```

```
Evaluation Steps:

1. Read the summary and the source document carefully.
2. Compare the summary to the source document and identify the main points of the article.
3. Assess how well the summary covers the main points of the article, and how much irrelevant
or redundant information it contains.
4. Assign a relevance score from 1 to 5.

Example:

Source Text:

{document}

Summary:

{summary}

Evaluation Form output the score only:
```

## A.3 PROMPTS USED FOR MTBENCH PREFERENCE DATA FOR SONNET

```
Human: For this task, you will be shown two conversations between a user and an AI assistant,
labeled A and B. Your goal is to evaluate which response (A or B) better follows the user's
instructions and more helpfully answers their question.

<PrefJudgment>
<Conversation A>
{conversation_a}
</Conversation A>

<Conversation B>
{conversation_b}
</Conversation B>

<Instructions>
Evaluate the two conversations and choose one of the following:
[[A]] if Conversation A's AI assistant better follows the user's instructions and answers
their question
[[B]] if Conversation B's AI assistant better follows the user's instructions and answers
their question
[[C]] if both AI assistants are equally good/poor in following instructions and answering the
user's question

Consider factors like helpfulness, relevance, accuracy, depth, creativity, and appropriate
level of detail when making your evaluation. Do not show positional bias towards A or B.
Response length should not unduly influence your decision.
Make a selection from "[[A]]", "[[B]]", "[[C]]". Only write the answer, do not write reasons.
</Instructions>
</PrefJudgment>

Assistant:
```

## A.4 PERCENTAGE AGREEMENT

**Percentage agreement:** The percentage agreement, as defined by McHugh (2012), for $n$ items – where $x_1, x_2, \ldots, x_n$ represent each item – that is labeled, with $R$ raters and $C$ categories, as the maximum percentage of votes the most frequent item gets as long as it is greater than 1. Formally:

$$\frac{1}{n} \sum_{i=1}^{n} \left( \max_{c \in \{1, \ldots, C\}} \left( \frac{1}{R} \sum_{r=1}^{R} \mathbb{1}_{\{x_i^r = c\}} \right) \cdot \mathbb{1}_{\left( \sum_{r=1}^{R} \mathbb{1}_{\{x_i^r = c\}} > 1 \right)} \right) \tag{3}$$

where $x_i^r$ is an annotation for $x_i$ by the $r$th rater. Note that the $\mathbb{1}_{\mathbf{A}}$ is the indicator function that returns 1 if condition $\mathbf{A}$ is true and 0 otherwise.

## A.5    IMPACT OF LABEL UNCERTAINTY ON ORDINAL SUMMEVAL DATASET

We create multiple subsets of 100 samples, with each subset having a predefined portion of samples with perfect agreement in human labels. We then randomly select samples from the dataset based on this portion. This simulation helps us understand how correlation measurements change with varying levels of human label uncertainty.

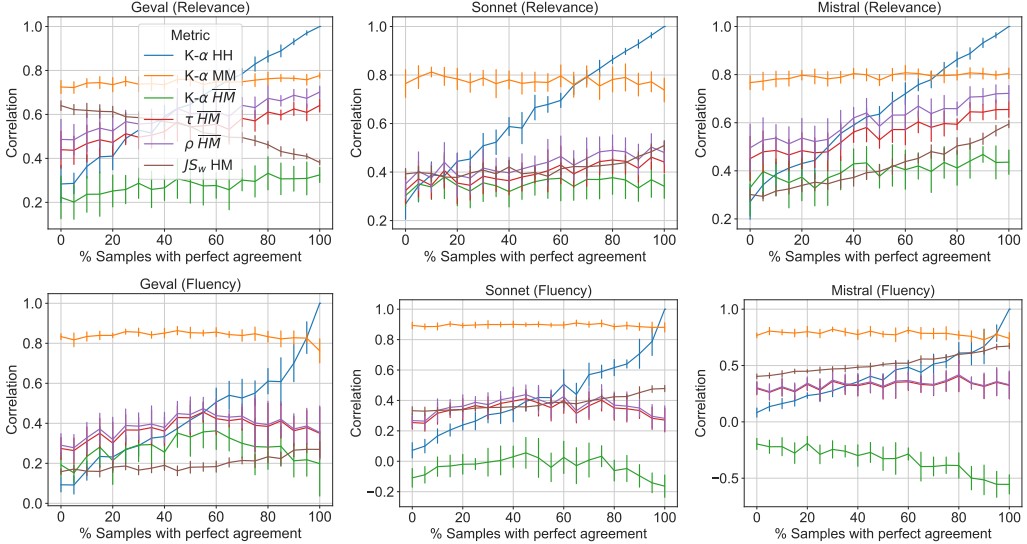

Figure 5: Impact of noise on ordinal dataset Summeval

## A.6    VISUALIZATION OF ROUGE-SCORE CORRELATION

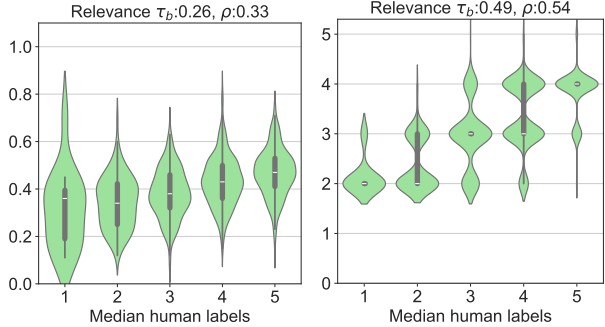

Figure 6: Visualizing correlation with human median rating for relevance: **(Left)** Rouge-1 **(Right)** LJ (Mistral). **Note:** Rouge scores are continuous values [0,1], humans and LJ use integer values 1-5.

### A.7 CODE FOR BINNED-JSD-PERCEPTION

Here we first show a working sample, followed by the code below.

In the sample dataset in Table 4, we have 2 bins, Bin 2 and Bin 3, where H represents humans and M represents model values

```
Values compared in Bin 2
                   = H(item A + item B),  M (item A + item B)
                   = H([2, 2, 3] + [1,2,2]), M([3,2] + [1,1])
                   = H([2,2,3,1,2,2]), M([3,2,1,1])

Values compared in Bin 3
                   = H(item  C), M( item C)
                   = H([2,3,3]), M([2, 2])
```

Now we compute JSD for each bin to compare the similarity in each bin. Comparing the probability distribution of values between H and M, assume Likert scale 1- 3, where the index represents the Likert value and the index value corresponds to the probability of that value:

```
Bin 2 JSD(H, M) = JSD(H[1/6, 4/6, 1/6], M[2/4, 1/4, 1/4])
Bin 3 JSD(H, M) = JSD(H[0, 1/3, 2/3], M[0, 2/2, 0])
```

Translating the above to python call

```
from scipy.spatial import distance
bin2_jsd = distance.jensenshannon([1/6, 4/6, 1/6], [2/4, 1/4, 1/4]))
bin3_jsd = distance.jensenshannon([0, 1/3, 2/3], [0, 2/2, 0])
```

Total binned JSD is the weighted sum of number of samples in each bin, where contains 2 samples A and B, contains 1 sample C

Binned JSD = 2/3 * $bin2_{jsd}$ + 1/3 * $bin3_{jsd}$ = 2/3 * 0.31 + 1/3 * 0.56 = 0.39

| Id | Human labels | Human median (Bin) | Model predictions |
|----|--------------|--------------------|--------------------|
| A  | 2, 2, 3      | 2                  | 3, 2               |
| B  | 1, 2, 2      | 2                  | 1, 1               |
| C  | 2, 3, 3      | 3                  | 2, 2               |

Table 4: Sample dataset

```
import itertools
import statistics
from collections import Counter
from statistics import mode
from scipy.spatial import distance

def compute_binned_js(df, human_labels_column, machine_labels_column, bin_type='median'):
    """
Compute JS
    @param df: DataFrame containing the columns human_labels_column, machine_labels_column

    @param human_labels_column: This is the name of the column containing human labels. Each
    row value in this column is a list of labels

    @param machine_labels_column: This is the name of the column containing machine labels.
    Each row value in this column is a list of multiple labels assigned by the machine

    @param bin_type: The type of binning, either median or majority depending on the type of
    data
    """
    result_weighted_js = 0
    result_bin_details = {}

    # Bin function can be median or mode ( most frequent value)
    bin_func = statistics.median if bin_type == 'median' else mode

    # Get unique labels
    labels = sorted(set(itertools.chain(*(df[human_labels_column].tolist()
                                      + df[machine_labels_column].tolist()))))
```

```
    # Assign bin number
    df_bin_number = df[human_labels_column].apply(lambda x: bin_func(x))

    for l in labels:

        # Get bin items
        df_bin_items = df[df_bin_number == l]

        bin_weight = len(df_bin_items) / len(df)

        result_bin_details[l] = {
            "bin_items": df_bin_items,
            "bin_number": l,
            "jsd": None,
            "bin_weight": bin_weight,
            "dist_human":[],
            "dist_machine":[]

        }

        # Skip if no items in bin
        if len(df_bin_items) == 0: continue

        # Compute js
        human_labels_frequencies_in_bin = Counter(
            list(itertools.chain(*df_bin_items[human_labels_column])))
        machine_labels_frequencies_in_bin = Counter(
            list(itertools.chain(*df_bin_items[machine_labels_column])))

        dist_human = [human_labels_frequencies_in_bin[i] / sum(Counter(
        human_labels_frequencies_in_bin).values())
                        for i in labels]

        dist_machine = [machine_labels_frequencies_in_bin[i] / sum(Counter(
        machine_labels_frequencies_in_bin).values())
                        for i in labels]

        result_bin_details[l]["dist_machine"]=dist_machine
        result_bin_details[l]["dist_human"]=dist_human

        jsd =  distance.jensenshannon(dist_human, dist_machine)
        result_bin_details[l]["jsd"] = jsd
        result_weighted_js += bin_weight * jsd

    return result_weighted_js, result_bin_details
```

## A.8   TOY EXAMPLE: APPLICATION OF BINNED-JSD AND WHERE IT CAN A BETTER METRIC

The use of binned-JSD solves a specific problem, where a single majority label is not sufficient to represent the human preference, as mentioned in section 4.2.

The advantage of binned-JSD can be demonstrated using a toy example as follows. If we only used a single gold human label (human median in this case, could also be human majority) to compute correlation, the acceptable values that humans have chosen is lost. As a result, metrics such as Krippendorff will see treat any value that is equidistant from the human single "gold" label as acceptably similar. For instance, assume that humans choose "disagree" or "neutral" (median/ majority value) (selections $\langle 2,3,3 \rangle$). A good model chooses "disagree" and a bad model chooses "agree" (completely different to human choice), because both "disagree" (Likert 2) and "agree" (Likert 4) are equidistant from the median/majority value (Likert 3- Median value), K-alpha has assigned very similar scores for the model that has assigned "disagree" (Likert 2) and the model that has assigned "agree" (Likert 4). Rank correlation metrics, in addition to their misfit in comparing item level Likert scores already discussed in the paper in section RQ2, also have a similar problem and deems the model that is poor as the better model as shown below. Our proposed approach, on the other hand, assigned lower (better for JSD) scores to the better model as the "good" model assigns values that the humans have chosen, compared to the poor model that has predicted a different score altogether.

The example above also exemplifies how unless we know apriori which is a better model, it is difficult to identify advantages as well as shortcomings with correlation measurements including the proposed binned-jsd. Effectiveness of metrics depend on the data. We don't know for certain if a model appears to be better/worse because of gaps in metrics, creating a chicken-and-egg problem

| Human labels | Human median ($\overline{H}$) | Good model | Poor model |
|---|---|---|---|
| 2, 2, 3 | 2 | 3 | 1 |
| 1, 2, 2 | 2 | 1 | 3 |
| 2, 3, 3 | 3 | 2 | 4 |

Table 5: A toy dataset to illustrate the use of binned-jsd

| Metric | Good Model | Poor Model | Does better model score better |
|---|---|---|---|
| Kendall-$\tau$ | 0.00 | 0.82 | False |
| Spearman-$\rho$ | 0.00 | 0.87 | False |
| Krippendorff-$\alpha$ | -0.06 | -0.06 | False |
| $JS_b$ | 0.56 | 0.65 | True |

Table 6: Demonstration how metrics can rank a poor model better as a consequence of ignoring valid labels that humans have assigned. The proposed Binned-JSD on the other hand uses all the human labels, and assigns lower JSD (better) to the better model.

in measuring the effectiveness of a metric itself. Not knowing which metric is appropriate is a common problem when it comes to correlation metrics (ten Hove et al., 2018), including problems with Cohen's Kappa (Krippendorff, 2004) despite it being commonly used, including many LLM as judge papers.

Hence, our recommendation in Section 4.3 for stratification, visualization and multi-metric reporting so we can interpret the strengths and gaps in the metrics.

| C | Partition (% samples) | Rouge1 $\rho$ | $\tau$ | Model | Krippendorff's-$\alpha$ HH | MM | $\overline{\text{HM}}$ | $\Delta$ | H'$\overline{\text{M}}$ | $JS_b$ HM | Spearman's-$\rho$ $\overline{\text{HM}}$ | H'$\overline{\text{M}}$ | H$^\mu$M$^\mu$ | Kendall's-$\tau$ $\overline{\text{HM}}$ | H'$\overline{\text{M}}$ |
|---|---|---|---|---|---|---|---|---|---|---|---|---|---|---|---|
| coherence | 1. All (100%) | 0.18 | 0.14 | G-Eval | 0.55 | 0.74 | 0.31 | 0.25 | 0.29 | 0.39 | 0.53 | 0.47 | 0.57 | 0.47 | 0.41 |
| | | | | Mistral | 0.55 | 0.78 | 0.49 | 0.07 | 0.43 | 0.36 | 0.56 | 0.49 | 0.62 | 0.49 | 0.43 |
| | | | | Sonnet | 0.55 | 0.83 | 0.24 | 0.32 | 0.22 | 0.51 | 0.34 | 0.33 | 0.41 | 0.30 | 0.29 |
| | 2. PA=1 (12%) | 0.36 | 0.28 | G-Eval | 1.00 | 0.78 | 0.25 | 0.75 | 0.25 | 0.70 | 0.61 | 0.61 | 0.64 | 0.55 | 0.55 |
| | | | | Mistral | 1.00 | 0.79 | 0.35 | 0.65 | 0.35 | 0.69 | 0.62 | 0.62 | 0.68 | 0.56 | 0.56 |
| | | | | Sonnet | 1.00 | 0.86 | 0.24 | 0.76 | 0.24 | 0.71 | 0.43 | 0.43 | 0.48 | 0.39 | 0.39 |
| | 3. $60 \leq$ PA $< 100\%$ (66%) | 0.16 | 0.12 | G-Eval | 0.53 | 0.75 | 0.30 | 0.23 | 0.29 | 0.40 | 0.51 | 0.45 | 0.55 | 0.45 | 0.39 |
| | | | | Mistral | 0.53 | 0.78 | 0.48 | 0.05 | 0.44 | 0.38 | 0.56 | 0.50 | 0.62 | 0.49 | 0.44 |
| | | | | Sonnet | 0.53 | 0.81 | 0.22 | 0.31 | 0.20 | 0.51 | 0.32 | 0.31 | 0.40 | 0.28 | 0.28 |
| | 4. PA $< 60\%$ (21%) | 0.17 | 0.13 | G-Eval | 0.19 | 0.70 | 0.32 | -0.13 | 0.26 | 0.35 | 0.56 | 0.42 | 0.58 | 0.52 | 0.38 |
| | | | | Mistral | 0.19 | 0.78 | 0.55 | -0.37 | 0.39 | 0.30 | 0.57 | 0.41 | 0.64 | 0.52 | 0.36 |
| | | | | Sonnet | 0.19 | 0.85 | 0.17 | 0.02 | 0.17 | 0.50 | 0.40 | 0.31 | 0.45 | 0.37 | 0.28 |
| relevance | 1. All (100%) | 0.33 | 0.26 | G-Eval | 0.40 | 0.74 | 0.25 | 0.15 | 0.25 | 0.37 | 0.53 | 0.47 | 0.59 | 0.47 | 0.41 |
| | | | | Mistral | 0.40 | 0.79 | 0.38 | 0.02 | 0.33 | 0.31 | 0.54 | 0.44 | 0.61 | 0.49 | 0.39 |
| | | | | Sonnet | 0.40 | 0.79 | 0.35 | 0.05 | 0.31 | 0.38 | 0.39 | 0.37 | 0.48 | 0.36 | 0.33 |
| | 2. PA (14%) | 0.38 | 0.30 | G-Eval | 1.00 | 0.76 | 0.30 | 0.70 | 0.30 | 0.63 | 0.69 | 0.69 | 0.73 | 0.63 | 0.63 |
| | | | | Mistral | 1.00 | 0.80 | 0.43 | 0.57 | 0.43 | 0.59 | 0.71 | 0.71 | 0.72 | 0.65 | 0.65 |
| | | | | Sonnet | 1.00 | 0.77 | 0.36 | 0.64 | 0.36 | 0.51 | 0.49 | 0.49 | 0.60 | 0.45 | 0.45 |
| | 3. $60 \leq$ PA $< 100\%$ (66%) | 0.34 | 0.26 | G-Eval | 0.40 | 0.74 | 0.25 | 0.16 | 0.25 | 0.38 | 0.53 | 0.47 | 0.60 | 0.47 | 0.41 |
| | | | | Mistral | 0.40 | 0.80 | 0.37 | 0.03 | 0.34 | 0.33 | 0.54 | 0.44 | 0.62 | 0.49 | 0.39 |
| | | | | Sonnet | 0.40 | 0.79 | 0.34 | 0.07 | 0.32 | 0.37 | 0.39 | 0.37 | 0.49 | 0.36 | 0.34 |
| | 4. PA $< 60\%$ (19%) | 0.28 | 0.22 | G-Eval | -0.09 | 0.69 | 0.18 | -0.27 | 0.22 | 0.33 | 0.40 | 0.31 | 0.42 | 0.38 | 0.27 |
| | | | | Mistral | -0.09 | 0.74 | 0.32 | -0.41 | 0.20 | 0.28 | 0.42 | 0.25 | 0.46 | 0.40 | 0.22 |
| | | | | Sonnet | -0.09 | 0.75 | 0.29 | -0.38 | 0.19 | 0.43 | 0.36 | 0.29 | 0.40 | 0.35 | 0.27 |

Table 7: All stratification results for SummEval. Summeval has 3 expert human annotations to select Likert values from 1-5, hence possible values for stratification by percentage agreement (PA) on the median label are 3/3=100 %, 2/3=66.67% and 1/3=33.3%. Hence, we stratify by 100% agreement, [60-100) and <60%.

## A.9 VISUALIZATION OF PREFERENCE DATA

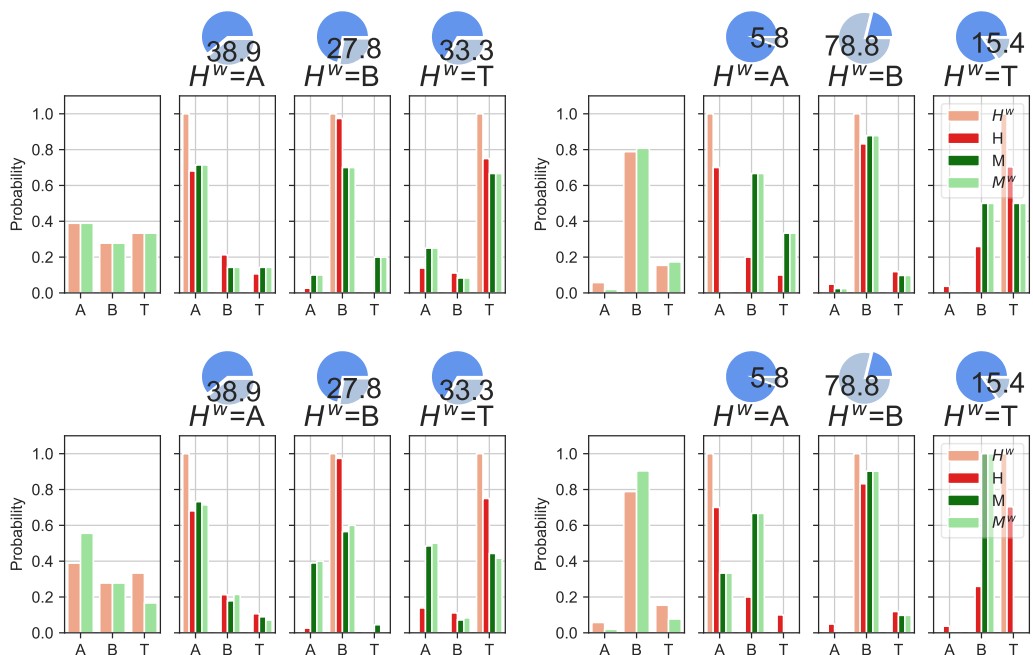

Figure 7: **Pairwise preference HM perception chart:** Human preference distribution vs. (*Top*) LJ GPT-4 (*Bottom*) LJ Claude Sonnet 3 for a given majority label per model pair. **T** is tie. Compares model pair (*Left*) Claude-v1 (**A**) vs. GPT3.5-turbo (**B**) (*Right*) Alpaca-13B (**A**) vs. GPT-3.5-turbo (**B**).

## A.10 EXTENDED TABLE RESULTS

| MP (A vs B) | LJ | Partition (# samples) | Human | | Krippendorff's-$\alpha$ HH | MM | $H^wM^w$ | $\Delta$ | $H'M^w$ | % Agreement HH | $H^wM^w$ | $H'M^w$ | $JS_b$ HM | Fleiss-$\kappa$ $H^wM^w$ | $H'M^w$ | Randolph-$\kappa$ $H^wM^w$ | $H'M^w$ | LJ Winrate W | WR | $\Delta$ |
|---|---|---|---|---|---|---|---|---|---|---|---|---|---|---|---|---|---|---|---|---|
| ⟨Alp, Gp3⟩ | GPT | 1. All (52) | B | 0.79 | 0.24 | - | 0.32 | -0.08 | 0.20 | 0.81 | 0.77 | 0.67 | 0.11 | 0.31 | 0.19 | 0.65 | 0.51 | B | 0.81 | 0.02 |
| | | 2. P-HH (26) | B | 0.92 | 1.00 | - | -0.05 | 1.05 | -0.05 | 1.00 | 0.85 | 0.85 | 0.13 | -0.07 | -0.07 | 0.77 | 0.77 | B | 0.92 | 0.00 |
| | | 3. HH: [80,100) (1) | B | 1.00 | 0.00 | - | - | - | 0.00 | 0.83 | 1.00 | 0.00 | 0.25 | - | -1.00 | - | -1.00 | B | 1.00 | 0.00 |
| | | 4. HH: [60,80) (18) | B | 0.61 | -0.04 | - | 0.36 | -0.40 | 0.21 | 0.68 | 0.67 | 0.56 | 0.22 | 0.34 | 0.18 | 0.50 | 0.33 | B | 0.67 | 0.06 |
| | | 5. HH: [0,60) (7) | B | 0.71 | -0.18 | - | 0.35 | -0.53 | -0.02 | 0.46 | 0.71 | 0.43 | 0.32 | 0.30 | -0.10 | 0.43 | 0.14 | B | 0.71 | 0.00 |
| | Sonnet | 1. All (52) | B | 0.79 | 0.24 | 0.90 | 0.01 | 0.23 | 0.09 | 0.81 | 0.73 | 0.71 | 0.18 | 0.00 | 0.08 | 0.60 | 0.57 | B | 0.90 | 0.12 |
| | | 2. P-HH (26) | B | 0.92 | 1.00 | 1.00 | -0.02 | 1.02 | -0.02 | 1.00 | 0.92 | 0.92 | 0.17 | -0.04 | -0.04 | 0.85 | 0.85 | B | 1.00 | 0.08 |
| | | 3. HH: [80,100) (1) | B | 1.00 | 0.00 | 0.00 | 0.00 | 0.00 | 0.00 | 0.83 | 0.00 | 0.00 | 0.36 | -1.00 | -1.00 | -1.00 | -1.00 | T | 1.00 | 0.00 |
| | | 4. HH: [60,80) (18) | B | 0.61 | -0.04 | 0.93 | -0.04 | 0.00 | 0.13 | 0.68 | 0.50 | 0.56 | 0.17 | -0.07 | 0.11 | 0.25 | 0.33 | B | 0.78 | 0.17 |
| | | 5. HH: [0,60) (7) | B | 0.71 | -0.18 | 1.00 | -0.08 | -0.10 | -0.18 | 0.46 | 0.71 | 0.43 | 0.51 | -0.17 | -0.27 | 0.43 | 0.14 | B | 1.00 | 0.29 |
| ⟨Cld, Gp3⟩ | GPT | 1. All (36) | A | 0.39 | 0.40 | - | 0.54 | -0.14 | 0.30 | 0.79 | 0.69 | 0.53 | 0.11 | 0.54 | 0.29 | 0.54 | 0.29 | A | 0.39 | 0.00 |
| | | 2. P-HH (14) | B | 0.64 | 1.00 | - | 0.25 | 0.75 | 0.25 | 1.00 | 0.57 | 0.57 | 0.10 | 0.22 | 0.22 | 0.36 | 0.36 | B | 0.57 | 0.07 |
| | | 3. HH: [80,100) (0) | - | - | - | - | - | - | - | - | - | - | 0.00 | - | - | - | - | - | - | - |
| | | 4. HH: [60,80) (21) | A | 0.52 | -0.02 | - | 0.58 | -0.60 | 0.28 | 0.67 | 0.76 | 0.52 | 0.12 | 0.57 | 0.26 | 0.64 | 0.29 | A | 0.52 | 0.00 |
| | | 5. HH: [0,60) (1) | A | 1.00 | 0.00 | - | - | - | 0.00 | 0.40 | 1.00 | 0.00 | 0.52 | - | -1.00 | - | -1.00 | A | 1.00 | 0.00 |
| | Sonnet | 1. All (36) | A | 0.39 | 0.40 | 0.78 | 0.36 | 0.05 | 0.18 | 0.79 | 0.58 | 0.47 | 0.17 | 0.35 | 0.16 | 0.38 | 0.21 | A | 0.56 | 0.17 |
| | | 2. P-HH (14) | B | 0.64 | 1.00 | 0.56 | -0.07 | 1.07 | -0.07 | 1.00 | 0.43 | 0.43 | 0.24 | -0.11 | -0.11 | 0.14 | 0.14 | B | 0.64 | 0.00 |
| | | 3. HH: [80,100) (0) | - | - | - | - | - | - | - | - | - | - | 0.00 | - | - | - | - | - | - | - |
| | | 4. HH: [60,80) (21) | A | 0.52 | -0.02 | 0.86 | 0.37 | -0.39 | 0.18 | 0.67 | 0.67 | 0.48 | 0.19 | 0.35 | 0.16 | 0.50 | 0.21 | A | 0.67 | 0.14 |
| | | 5. HH: [0,60) (1) | A | 1.00 | 0.00 | 1.00 | - | - | - | 0.40 | 1.00 | 1.00 | 0.52 | - | - | - | - | - | A | 1.00 | 0.00 |
| ⟨Gp3, Gp4⟩ | GPT | 1. All (38) | B | 0.53 | 0.26 | - | 0.27 | -0.01 | 0.11 | 0.74 | 0.61 | 0.50 | 0.18 | 0.26 | 0.10 | 0.41 | 0.25 | B | 0.74 | 0.21 |
| | | 2. P-HH (11) | B | 0.82 | 1.00 | - | 0.64 | 0.36 | 0.64 | 1.00 | 0.91 | 0.91 | 0.16 | 0.63 | 0.63 | 0.86 | 0.86 | B | 0.91 | 0.09 |
| | | 3. HH: [80,100) (2) | T | 1.00 | -0.12 | - | 0.00 | -0.12 | 0.00 | 0.80 | 0.50 | 0.50 | 0.23 | -0.33 | -0.33 | 0.00 | 0.00 | B | 0.50 | 0.50 |
| | | 4. HH: [60,80) (20) | B | 0.50 | 0.08 | - | 0.24 | -0.17 | -0.12 | 0.68 | 0.55 | 0.30 | 0.21 | 0.22 | -0.15 | 0.33 | -0.05 | B | 0.65 | 0.15 |
| | | 5. HH: [0,60) (5) | A | 0.60 | -0.27 | - | -0.24 | -0.02 | -0.08 | 0.40 | 0.20 | 0.40 | 0.38 | -0.38 | -0.20 | -0.20 | -0.20 | B | 0.80 | 0.20 |
| | Sonnet | 1. All (38) | B | 0.53 | 0.26 | 0.76 | 0.20 | 0.06 | 0.32 | 0.74 | 0.50 | 0.58 | 0.13 | 0.19 | 0.31 | 0.25 | 0.37 | B | 0.45 | 0.08 |
| | | 2. P-HH (11) | B | 0.82 | 1.00 | 0.66 | 0.19 | 0.81 | 0.19 | 1.00 | 0.64 | 0.64 | 0.28 | 0.15 | 0.15 | 0.45 | 0.45 | B | 0.64 | 0.18 |
| | | 3. HH: [80,100) (2) | T | 1.00 | -0.12 | 0.12 | - | - | - | 0.80 | 1.00 | 1.00 | 0.07 | - | - | - | - | T | 1.00 | 0.00 |
| | | 4. HH: [60,80) (20) | B | 0.50 | 0.08 | 0.81 | 0.22 | -0.15 | 0.46 | 0.68 | 0.50 | 0.65 | 0.06 | 0.20 | 0.44 | 0.25 | 0.48 | B | 0.45 | 0.05 |
| | | 5. HH: [0,60) (5) | A | 0.60 | -0.27 | 0.64 | -0.45 | 0.19 | -0.41 | 0.40 | 0.00 | 0.00 | 0.49 | -0.61 | -0.56 | -0.50 | -0.50 | T | 0.80 | 0.20 |

Table 8: All stratification results for MTBench. MTBench has varying number of 3 to 5 human annotations per item to select preferences A, B or tie. Hence, possible values for stratification by percentage agreement (PA), so if an item contains 5 annotations (human labels), then possible stratification groups that the item can belong to by PA on the majority label are 5/5=100%, 4/5=80%, 3/5=60% or lesser. Please note that many of these partitions have 0 or less than 5 samples. With higher proportion of noisy or uncertain samples the HM correlation seems to outperform HH correlation. The exceptions to this finding are highlighted, and for these the sample size is quite small ($\leq 5$).

## A.11  RANDOM SIMULATION CODE

### A.11.1  RANDOM SIMULATION CODE - BINARY CLASSIFICATION (NOMINAL DATA)

In this code, we create a simulated synthetic dataset, where we mimic the case of 2 human annotators labelling a binary classification problem. We also create a random labeller, where the random labeller assigns random labels.

```python
import random

import numpy as np
import random
import pandas as pd

def synthetic_random_nominal_dataset():
    """
    Simulates a random binary dataset with 2 human labellers.
    The 2 humans can either (a) both pick 0 (b)both pick 1 (c) one picks 0 and the other picks
     1 or vice versa
    :return:
    """
    dataset_size = 200
    humans_2_one_pick_0_other_picks_1 = [random.sample([0, 1], 2) for _ in range(dataset_size
    // 2)]
    humans_2_both_pick_1 = [[1, 1] for _ in range(dataset_size // 4)]
    humans_2_both_pick_0 = [[0, 0] for _ in range(dataset_size // 4)]

    human_2_annotators_binary_simulated = humans_2_one_pick_0_other_picks_1 +
    humans_2_both_pick_1 + humans_2_both_pick_0

    random_labeller_choice = [np.random.choice([1, 0]) for _ in range(dataset_size)]

    # Final df
    df = pd.DataFrame(data={"human_labels": human_2_annotators_binary_simulated,
                            "random_labeller": random_labeller_choice
                            })

    return df
```

### A.11.2  RANDOM SIMULATION CODE - 3 WAY CLASSIFICATION (ORDINAL DATA)

In this code, we create a simulated synthetic dataset, where we mimic the case of 2 human annotators labelling a 3 way classification problem to assign labels [1,2 or 3]. We also create a random labeller, where the random labeller assigns random labels.

```python
import numpy as np
import random
import pandas as pd

def synthetic_random_ordinal_dataset():
    """
    Simulates a random 3 way classification 1-2-3, with 2 human labellers.
    The 2 humans can either (a) both pick 1 (b)both pick 2. and so on (c) disagree
    :return:
    """
    dataset_size = 600
    humans_disagree = [random.sample([1, 2, 3], 2) for _ in range(dataset_size // 2)]
    humans_agree_1 = [[1, 1] for _ in range(dataset_size // 6)]
    humans_agree_2 = [[2, 2] for _ in range(dataset_size // 6)]
    humans_agree_3 = [[3, 3] for _ in range(dataset_size // 6)]

    human_2_annotators_ordinal_simulated = humans_disagree + humans_agree_1 + humans_agree_2 +
     humans_agree_3

    random_labeller_choice = [np.random.choice([1, 2, 3]) for _ in range(dataset_size)]

    df = pd.DataFrame(data={"human_labels": human_2_annotators_ordinal_simulated,
                            "random_labeller": random_labeller_choice
                            })

    return df
```

## A.12 CASE STUDY

Here, we include a case study on a subset of the datasets (see details in Table 9) used in the LLM benchmark study by Bavaresco et al. (2024). We use the preprocessed datasets available in github[1] provided by Bavaresco et al. (2024) These datasets were evaluated on LLM Llama-3-70B.

| Dataset | Size | Criteria | # Annotations | Choices |
|---|---|---|---|---|
| Topical chat (Mehri & Eskenazi, 2020) | 60 | Understandable | 3 | (0, 1) |
| | 60 | Uses Knowledge | 3 | (0, 1) |
| QAGS (Wang et al., 2020) | 953 | Factual consistency | 3 | (yes, no) |
| Dices 350 (Aroyo et al., 2023) | 350 | Safety | 100-150 | (yes, no, unsure) |

Table 9: Categorical datasets summary

| D | C | Partition | Size (%) | Krippendorff-$\alpha$ | | | | % Agreement | | | | Randolph-$\kappa$ | | | |
|---|---|---|---|---|---|---|---|---|---|---|---|---|---|---|---|
| | | | | HH | MM | H$^w$M$^w$ | $\Delta$ | HH | MM | H$^w$M$^w$ | $\Delta$ | HH | MM | H$^w$M$^w$ | $\Delta$ |
| TC | UND | All | 100.0 | -0.01 | 0.58 | -0.01 | 0.00 | 0.99 | 0.98 | 0.97 | 0.02 | 0.96 | 0.94 | 0.93 | 0.02 |
| | | PA=1 | 96.7 | 1.00 | 0.58 | -0.01 | 1.01 | 1.00 | 0.98 | 0.97 | 0.03 | - | 0.94 | 0.93 | - |
| | | PA=66.67% | 3.3 | -0.25 | 1.00 | - | - | 0.67 | 1.00 | 1.00 | -0.33 | -0.33 | - | - | - |
| | UK | All | 100.0 | 0.48 | 0.81 | -0.03 | 0.51 | 0.98 | 0.99 | 0.93 | 0.04 | 0.91 | 0.98 | 0.87 | 0.04 |
| | | PA=1 | 93.3 | 1.00 | 1.00 | -0.01 | 1.01 | 1.00 | 1.00 | 0.96 | 0.04 | 1.00 | 1.00 | 0.93 | 0.07 |
| | | PA=66.67% | 6.7 | -0.26 | 0.44 | -0.17 | -0.09 | 0.67 | 0.90 | 0.50 | 0.17 | -0.33 | 0.70 | 0.00 | -0.33 |
| QG | FC | All | 100.0 | 0.49 | 0.93 | 0.70 | -0.21 | 0.89 | 0.98 | 0.87 | 0.02 | 0.54 | 0.94 | 0.74 | -0.20 |
| | | PA=1 | 65.6 | 1.00 | 0.94 | 0.84 | 0.16 | 1.00 | 0.99 | 0.94 | 0.06 | 1.00 | 0.95 | 0.87 | 0.13 |
| | | PA=66.67% | 34.4 | -0.34 | 0.91 | 0.48 | -0.81 | 0.67 | 0.98 | 0.74 | -0.08 | -0.33 | 0.91 | 0.49 | -0.82 |
| DC | SAF | All | 100.00 | 0.16 | 0.79 | -0.25 | 0.41 | 0.69 | 0.93 | 0.39 | 0.30 | 0.35 | 0.82 | 0.09 | 0.26 |
| | | PA = 1 | 0.00 | - | - | - | - | - | - | - | - | - | - | - | - |
| | | 80% <= PA<100% | 22.6 | 0.31 | 0.81 | -0.41 | 0.72 | 0.86 | 0.94 | 0.34 | 0.52 | 0.63 | 0.85 | 0.01 | 0.62 |
| | | 60% <= PA <80% | 48.6 | 0.15 | 0.80 | -0.27 | 0.42 | 0.71 | 0.93 | 0.39 | 0.32 | 0.34 | 0.83 | 0.08 | 0.26 |
| | | 40% <= PA<60% | 28.9 | 0.02 | 0.76 | -0.11 | 0.12 | 0.53 | 0.92 | 0.43 | 0.10 | 0.15 | 0.79 | 0.14 | 0.01 |

Table 10: Categorical datasets evaluation: Results on Topical chat (TC) on criteria ⟨Understandable (UND), Uses Knowledge (UK)⟩, QAGS (QA) on criteria factual consistency (FC) and DICES (DC) dataset on safety (SAF). With higher proportion of noisy or uncertain samples the HM correlation seems to outperform HH correlation. The only one exception is highlighted, where the sample size is quite small ($\approx$ 4 samples as TC dataset has a total of 60 samples) in partition PA=66.67% (2/3 votes for majority label).

### A.12.1 ANALYSIS OF RESULTS

1. **Effects of Multi-metric reporting:** On the Topical Chat (TC) dataset, for understandable criteria the aggregate number (Krippendorff-$\alpha$ -0.01) of H$^w$M$^w$ score of -0.01 *superficially seems to imply* that HM correlation is low as shown in Table 10. However, percentage agreement (score 0.97) and Randolph-$\kappa$ (score 0.93) score quite highly, indicating that class imbalance has substantially lowered Krippendorff-$\alpha$ pretty close to 0.0. Also note that over 96% of the samples have perfect human agreement, however the overall HH Krippendorff-$\alpha$ is quite low scoring -0.01. This effect of how various chance adjusted metrics impact correlation scores is also discussed in detail in section 4.2.

2. **Impact of stratification:** When we compare the overall performance (column All in Table 10) on dataset TC (criteria understandable) with dataset QAGS, Randolph-$\kappa$ drops substantially by 19 points (0.93 → 0.74). However, QAGS dataset has around 66% of the samples that have perfect human agreement, while TC has 96% of the sample with human agreement. When we compare the samples with perfect human agreement between the 2 datasets, the model performance gap reduces to just 6 points (0.93 → 0.87), pointing to how comparing datasets with different proportion of uncertain samples can affect our conclusion (in this case *incorrectly that the model performance is substantially lower in QAGS compared to TC*). With DICES dataset (crowdsourced with over 100 annotations per item, with no perfect agreement items), on the other hand, the model seems to struggle across

---
[1]https://github.com/dmg-illc/JUDGE-BENCH

all metrics and stratification groups, indicating much deeper investigation is required. The general trend of where in a stratified group with relatively higher proportion of uncertain samples (as measured by low HH correlation), the $H^w M^w$ correlation seems to outperform HH correlation as highlighted, as shown in Table 10 also applies, indicating to how *models can superficially appear to approximate human majority when proportion samples of uncertainty is relatively higher*, demonstrating the impact of RQ1 discussed in section 3.1.

### A.12.2 ANALYSIS OF ORDINAL DATA

Firstly, looking at Table 11, superficially looking at the aggregate scores ( Row marked "All" across all 4 criteria) it might appear as though **(a)** absolute human to machine correlation is low (columns $\overline{\text{HM}}$) **(b)** when we compare HM correlation to HH, $\overline{\text{HM}}$ is similar to HH correlation.

| Dataset | Size | Criteria | # Annotations | Choices |
|---|---|---|---|---|
| | 60 | engaging | 3 | 1,2,3 |
| | 60 | maintains context | 3 | 1,2,3 |
| Topical chat (Mehri & Eskenazi, 2020) | 60 | natural | 3 | 1,2,3 |
| | 60 | overall | 3 | 1,2,3,4,5 |

| | | | Krippendorff's-$\alpha$ | | | | $JS_b$ | $\rho$ | $\tau$ |
|---|---|---|---|---|---|---|---|---|---|
| C | Partition | Size (%) | HH | MM | $\overline{\text{HM}}$ | $\Delta$ | HM | $\overline{\text{HM}}$ | $\overline{\text{HM}}$ |
| ENG | All | 100.00 | 0.03 | 0.72 | -0.02 | 0.05 | 0.24 | 0.25 | 0.24 |
| | PA=1 | 75.00 | 1.00 | 0.66 | -0.14 | 1.14 | 0.34 | - | - |
| | 60% $\leq$ PA < 80% | 25.00 | -0.39 | 0.82 | -0.01 | -0.38 | 0.15 | 0.26 | 0.25 |
| MC | All | 100.00 | -0.06 | 0.74 | -0.07 | 0.01 | 0.13 | - | - |
| | PA=1 | 81.67 | 1.00 | 0.65 | -0.04 | 1.04 | 0.20 | - | - |
| | 60% $\leq$ PA < 80% | 18.33 | -0.45 | 0.78 | -0.17 | -0.28 | 0.07 | - | - |
| NAT | All | 100.00 | -0.07 | 0.71 | -0.20 | 0.13 | 0.24 | - | - |
| | PA=1 | 78.33 | 1.00 | 0.75 | -0.18 | 1.18 | 0.35 | - | - |
| | 60% $\leq$ PA < 80% | 21.67 | -0.46 | 0.59 | -0.25 | -0.21 | 0.14 | - | - |
| OV | All | 100.00 | 0.10 | 0.75 | -0.28 | 0.38 | 0.33 | -0.17 | -0.16 |
| | PA=1 | 61.67 | 1.00 | 0.76 | -0.21 | 1.21 | 0.41 | - | - |
| | 60% $\leq$ PA < 80% | 33.33 | -0.31 | 0.72 | -0.48 | 0.17 | 0.35 | -0.37 | -0.35 |
| | 0 $\leq$ PA < 40% | 5.00 | -0.32 | 0.66 | -0.25 | -0.07 | 0.24 | - | - |

Table 11: Analysis of performance on the Topical chat dataset on criteria Engaging (**ENG**), Maintains context (**MC**), Naturalness (**NAT**) and Overall (**OV**). Overall, all the metrics that measure human – to machine correlation, across all metrics, seem quite low. At this point, it is quite difficult to comprehend what is causing this, until we visualize them as shown in Figures 8 – Figures11

In order to investigate the poor $\overline{\text{HM}}$, we perform the following analysis.

1. **Is the $\overline{\text{HM}}$ low because the model predictions are indeed quite different to humans?** Investigating the criteria Engaging, as shown in Table 11, the $\overline{\text{HM}}$ correlation is low (Krippendorff-$\alpha$ -0.02, $\rho$ $\overline{0.25}$, and $\tau$ 0.24). When we visualize this data, we find that over 96% of the human labels have a median of 3, and only 3.3% (2 out of 60) have a different median value ($\overline{\text{H}} = 2$), as shown in Figure 8. Obviously, there is a considerable class imbalance, that is *likely* to lower Krippendorff-$\alpha$. In the case of rank correlation, over 96% of $\overline{\text{H}} = 3$, and of these roughly 30% have been assigned a score of 3 $\overline{\text{M}} = 3$, lowering rank correlation metrics.

   Given this imbalance, there are 2 scenarios – **(a)** human annotation is reliable and the LLM prompts need to be optimized to improve the performance **(b)** Human annotation

guidelines need to be revised. If we **assume that the human annotation is reliable**, then the actionable step here is to ensure that the LLM prompt is optimized such that the items are incorrectly labelled as 2, are correct to 3.

We can see a similar pattern in criteria MC (see Figure 9), NAT (see Figure 10). For criteria OV, in LJ labels are quite different to machine labels (see Figure 11).

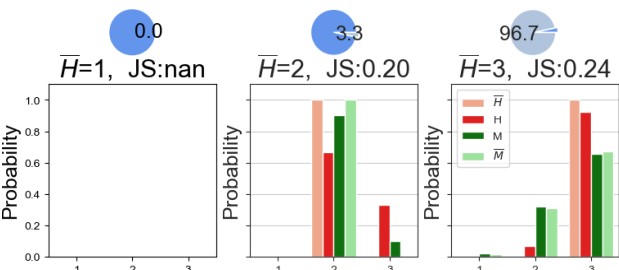

Figure 8: Topical chat criteria – Engaging: Over 96% of the samples are assigned a median score of 3 by humans fairly consistently, where around 90% of the human label bin $\overline{H} = 3$ have the same value as the median. Only 2 (3.3% of 60) items are in bin $\overline{H} = 2$. There are no human labels for ordinal value 1, as seen in bins 1, 2 and 3. This shows how unbalanced the human labels are.

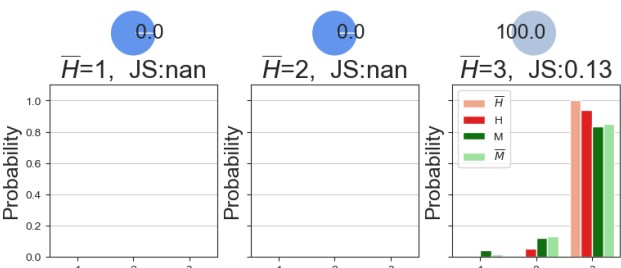

Figure 9: Topical chat criteria – Maintains context. The human median is 3 for 100% of the samples. Hence, rank correlation is not applicable . While in majority of the case (80% of the samples), human median seems to match the machine median, k-$\alpha$ correlation $\overline{HM}$ is quite low at -0.07 as shown Table 11, pointing to class imbalance as less than $< 10\%$ samples are assigned 2 as the median LJ rating.

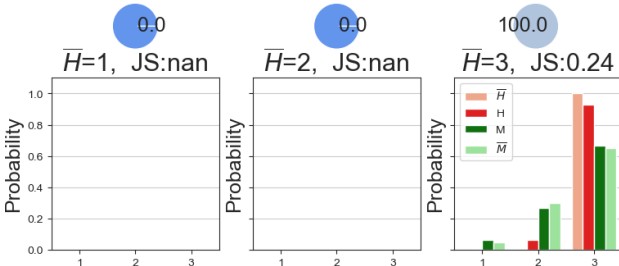

Figure 10: Topical chat criteria – Natural. The human median is 3 for 100% of the samples. While here approximately 60% of the samples are also labelled 3 by LJ, the k-$\alpha$ correlation $\overline{HM}$ is substantially low at -0.20 as shown Table 11.

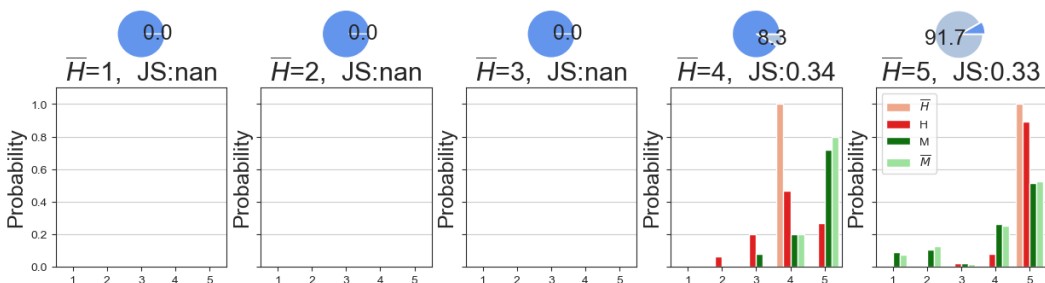

Figure 11: Topical chat criteria – Overall. Here the human labels for bin $\overline{H}$=4 vary, where only 40% of the labels match the median of 4, hence humans are uncertain about this label assignment. Also note, this bin $\overline{H}$=4 applies to 8.3% (5/60) of the samples. For the rest of 91% of the samples, human uncertainty is low, but LJ median labels $\overline{M}$ vary are quite different to human median $\overline{H} = 5$.

## A.13    EXAMPLES COMPARING HUMANS AND MACHINE LABELS

Here are some examples from SNLI datasets, where humans have perfect agreement (5/5 votes for the majority label), but LLM (LLama3.1 8B) predicts a different label as shown in Table 12. In Table 13, we show examples of where humans disagree on the labels, and where the majority label differs between human and machine.

---

**Premise:** Two women are embracing while holding to go packages.
**Hypothesis:** The men are fighting outside a deli.
**Human Labels:** contradiction; contradiction; contradiction; contradiction; contradiction
**Machine Labels:** neutral; neutral; neutral; neutral; neutral

---

**Premise:** Two young children in blue jerseys, one with the number 9 and one with the number 2 are standing on wooden steps in a bathroom and washing their hands in a sink.
**Hypothesis:** Two kids in jackets walk to school.
**Human Labels:** contradiction; contradiction; contradiction; contradiction; contradiction
**Machine Labels:** neutral; neutral; neutral; neutral; neutral

---

**Premise:** A brown a dog and a black dog in the edge of the ocean with a wave under them boats are on the water in the background.
**Hypothesis:** The dogs are swimming among the boats.
**Human Labels:** entailment; entailment; entailment; entailment; entailment
**Machine Labels:** neutral; neutral; neutral; neutral; neutral

---

**Premise:** A group of people prepare hot air balloons for takeoff.
**Hypothesis:** There are hot air balloons on the ground and air.
**Human Labels:** neutral; neutral; neutral; neutral; neutral
**Machine Labels:** entailment; entailment; entailment; entailment; entailment

---

**Premise:** A young child is jumping into the arms of a woman wearing a black swimming suit while in a pool.
**Hypothesis:** Mother catching her son in a pool.
**Human Labels:** neutral; neutral; neutral; neutral; neutral
**Machine Labels:** entailment; entailment; entailment; entailment; entailment

---

**Premise:** Three women in dress suits walk by a building.
**Hypothesis:** Three women are traveling by foot.
**Human Labels:** entailment; entailment; entailment; entailment; entailment
**Machine Labels:** neutral; neutral; neutral; neutral; neutral

---

Table 12: Examples in SNLI dataset where humans have perfect agreement, but the model produces a different answer.

| |
|---|
| **Premise:** Families waiting in line at an amusement park for their turn to ride. 
 **Hypothesis:** People are waiting in line at a restaurant. 
 **Human Labels:** entailment; contradiction; entailment; contradiction; contradiction 
 **Machine Labels:** neutral; neutral; neutral; neutral; neutral |
| **Premise:** A woman is writing something on a post-it note which is hanging on a bulletin board with a lot of other post-it notes. 
 **Hypothesis:** The woman is talking on the phone. 
 **Human Labels:** contradiction; contradiction; contradiction; entailment; contradiction 
 **Machine Labels:** neutral; neutral; neutral; neutral; neutral |
| **Premise:** A young dark-haired woman crouches on the banks of a river while washing dishes. 
 **Hypothesis:** A woman washes dishes in the river while camping. 
 **Human Labels:** neutral; contradiction; neutral; neutral; neutral 
 **Machine Labels:** entailment; entailment; entailment; entailment; entailment |
| **Premise:** A youth is kicking a soccer ball in an empty brick area. 
 **Hypothesis:** A funny human kicking. 
 **Human Labels:** neutral; neutral; neutral; contradiction; neutral 
 **Machine Labels:** entailment; entailment; entailment; entailment; entailment |
| **Premise:** A golfer has just finished swinging his club. 
 **Hypothesis:** a human outside 
 **Human Labels:** entailment; entailment; entailment; entailment; contradiction 
 **Machine Labels:** neutral; neutral; neutral; neutral; neutral |
| **Premise:** A woman in black, seen from behind, sits next to a body of water. 
 **Hypothesis:** A woman sits outside. 
 **Human Labels:** entailment; contradiction; entailment; entailment; entailment 
 **Machine Labels:** neutral; neutral; neutral; neutral; neutral |

Table 13: Examples of humans annotations in SNLI datasets where humans disagree, and the majority machine label is a different answer.

