# OpenReview forum: "Beyond correlation: The impact of human uncertainty in measuring the effectiveness of automatic evaluation and LLM-as-a-judge"
_ICLR.cc/2025/Conference — ICLR 2025 Poster_

### Official Review · Reviewer_RFAo · 2024-10-26

**Soundness:** 2
**Presentation:** 3
**Contribution:** 2
**Rating:** 5
**Confidence:** 5

**Summary:**

This paper examines how human uncertainty affects evaluating generative models, noting that standard metrics like Krippendorff’s α may misrepresent machine accuracy when human judgments vary. The authors propose three main contributions: stratifying evaluation results by human uncertainty levels, introducing binned Jensen-Shannon Divergence (JSb) to better measure alignment with human perception, and creating perception charts to visualize evaluation performance more effectively. These tools aim to provide a clearer, more accurate picture of machine evaluation performance amidst human variability.

**Strengths:**

Originality: This paper takes an original approach by addressing human uncertainty in generative model evaluation. Introducing stratified results by human variability and the new JSb metric for perception-based tasks adds fresh methods to handle subjectivity in evaluations. The perception charts also offer an innovative way to visualize nuanced performance differences.

Quality: The work is methodologically sound, with comprehensive experiments across datasets like SummEval and MNLI. The use of both real and synthetic data strengthens the empirical basis, showcasing the impact of human judgment noise on evaluation reliability.

Clarity: The paper is well-organized, clearly defining key concepts such as HH vs. HM correlation. Explanations of JSb and perception charts are straightforward, helping readers understand the new evaluation tools effectively.

Significance: This work fills an important gap by addressing subjective variability in human evaluations. Its proposed methods (if widely adopted) could lead to more accurate and relevant model evaluations, especially in perception-driven tasks across AI.

**Weaknesses:**

- The choice to stratify results based on “high” and “low” human uncertainty needs clearer justification. A discussion or empirical test on how these thresholds were set would make the stratification process more robust and reproducible.

- The introduction of Jensen-Shannon Divergence as a measure of perception-based tasks is promising, but its effectiveness is not thoroughly proven. Including a comparison with other potential metrics, such as Earth Mover’s Distance or Wasserstein Distance, would better validate the claim that JSb captures human perception more accurately.

-While reusing prior prompts is convenient, this may introduce biases or inconsistencies across models. Optimizing prompts specifically for each model would yield more accurate comparisons, especially given the importance of prompt sensitivity in LLM performance. Adding prompt-tuning experiments for each model could further solidify the findings.

-The synthetic data used to simulate high uncertainty scenarios lacks details on its generation process. More transparency on how closely this data reflects real-world scenarios, including its validation process, would help verify the relevance of the findings.

- The shifts in ∆ (difference between HH and HM correlations) across different uncertainty levels are intriguing but underexplored. An in-depth analysis of these shifts—perhaps with concrete examples of where machines diverge from human judgment—would help understand the implications of these results more clearly. Visual examples showing alignment and divergence between human and machine judgments could greatly enhance interpretability.

- The experiments rely primarily on four datasets, which, while varied, still represent a narrow slice of possible applications.

**Questions:**

- How do you decide what's "high" or "low" uncertainty in your stratification, and did you try other thresholds?
- Why did you choose Jensen-Shannon Divergence for human perception, and can you show this is better than existing metrics?
- How did you adapt Krippendorff’s α and similar metrics to account for systematic machine errors, not just random ones?
- Why use previous prompts without optimizing them for each model, wouldn't  this affect fairness in comparisons?
- Can you explain how you created the synthetic data for high uncertainty and whether it really reflects real-world data?
- Could you provide concrete examples where machine evaluations diverge significantly from human judgments?
- How would these findings help practitioners improve real-world model evaluations?

---

> ### Author Response · Authors · 2024-11-22
> **Reviewer 4 - W1**
>
> 1. The choice to stratify results based on “high” and “low” human uncertainty needs clearer justification. A discussion or empirical test on how these thresholds were set would make the stratification process more robust and reproducible.
>
> Thank you for your comments to help improve the clarification. We had used “high” and “low” certainty in the context of the random labeller as a shorthand to indicate relatively high proportions of samples with uncertainty, as we have figures that were plotted against varying proportions  samples with  uncertain labels.
>
>  In hindsight, we now understand the source of confusion. We have updated the content as follows under figure 2  as follows
> *Simulating the impact of uncertainty  by comparing with an automatic random  labeler **(R)**. When the proportion of samples with  uncertainty is higher, even a random labeler can \textit{appear} to have better correlation with a majority ($\text{H}^w$) or median {$\overline{H}$} human label”*
>
> We also updated the content under RQ1 to clarify high and low better  (referring to the figures where until a cut off is reached the random labeller appears better) - *In this scenario,  even a random labeler can appear better when the  proportion of samples with uncertainty is higher. It is only when the proportion of samples with consistent  labels increase, does the relative poor performance of the random labeler come to light,  shown in Fig.2. Intuitively,  if 2 humans disagree, a 3rd random labeler cannot do any worse in-terms of agreement. The random labeler can only disagree (in which case they are no better than the 2 humans)  or agree with any chosen  label. \textbf{This example further illustrates why stratification by proportion of uncertainty  is crucial to uncovering weaknesses in \textit{any automatic labeler}*
>
>
>
> In terms of deciding the stratification thresholds for the datasets, here are the details.
>
>
>
> In Table 1, the MNLI and SNLI datasets have exactly 5 human labels per item. Hence, you have the following scenarios – 5/5 = 1.0, 4/5 = 0.8, 3/5=0.6. Hence, we have stratified by all possible votes for the majority label.  In Table 2, for SummEval, there are exactly 3 human labels. Hence, you have 3/3=1.0, 2/3=0.67, 1/3 =0.33, for median label.   In Table 2, we only showed the max agreement and least agreement for brevity, we have now included all the stratification in Appendix In Table 5 and explained the stratification. The results do not change and remain the same.
>
>
>
> For Table 3, Each item  has 3-5 human label (varying number of human labels), where majority of the items have closer to 3 labels. The main challenge with the MTBench dataset displayed in table 2, is the sample size is quite small for each model pair with multiple annotations. hence some partitions have less than 5 samples or even 0, unlike the MNLI or SNLI datasets which have a few thousand samples, Hence, we have reported the partitions with resonable number of samples, which is 100\%, 60-80. We have now included all the partitions with even 1  sample in Appendix in Table 6. Our  findings hold, across all partitions, in table 6 as well .

---

> ### Author Response · Authors · 2024-11-22
> **Reviewer 4 - W2**
>
> 2. jensen-Shannon Divergence as a measure of perception-based tasks is promising, but its effectiveness is not thoroughly proven.
>
> Thank you for your question. Various reviewers have asked the same question.
>
> Here is our explanation.
> The use of binned-JSD solves a specific problem, where a single majority label is not sufficient to represent the human preference, as mentioned in section 4.2.
>
> Consider the example in the table below. If only used a single gold human label (human_median in this case, could also be human_majority) to compute correlation, the acceptable values that humans have chosen is lost. As a result, metrics such as Krippendorff will see treat any value that is equidistant from the human single “gold” label as acceptably similar. For instance, assume that humans choose “disagree” or “neutral” (median/ majority value) (selections <2,3,3>). A good model chooses ''disagree" and a bad model chooses “agree” (completely different to human choice), because both “disagree” (Likert 2) and “agree” (Likert 4) are equidistant from the median/majority value (Likert 3- Median value), K-alpha has assigned very similar scores for the model that has assigned “disagree” (Likert 2) and the model that has assigned “agree” (Likert 4). Rank correlation metrics, in addition to their misfit in comparing item level Likert scores already discussed in the paper in section RQ2, also have a similar problem and deems the model that is poor as the better model as shown below. Our proposed approach on the other hand, assigned lower (better for JSD) scores to the better model as the "good” model assigns values that the humans have chosen, compared to the poor model that has predicted a different score altogether.
>
> | humans    |   human_median |   model_good |   model_poor |
> |:----------|---------------:|-------------:|-------------:|
> | [2, 2, 3] |              2 |            3 |            1 |
> | [1, 2, 2] |              2 |            1 |            3 |
> | [2, 3, 3] |              3 |            2 |            4 |
>
> Metric results
> | metric   |   model_good |   model_poor | Does better model score better   |
> |----------|--------------|--------------|----------------------------------|
> | Tau      |         0    |         0.82 | False                            |
> | Rho      |         0    |         0.87 | False                            |
> | K-alpha  |        -0.06 |        -0.06 | False                            |
> | JS_b     |         0.56 |         0.65 | True                             |
>
>
>  **Note** Lower JSD (more similar hence indicates better model) as JSD is a divergence metric, like distance. In addition, ranges between 0 (min) and 1 max, for log to the base 2 as indicated in explanation for equation 2.
>
> The example above also exemplifies how unless we know apriori which is a better model, it is difficult to identify advantages / shortcomings with correlation measurements including the proposed binned-jsd. Effectiveness of metrics depend on the data. We dont know for certain if a model appears to be better/worse because of gaps in metrics, creating a chicken and egg problem in measuring the effectiveness of a metric itself. Not knowing which metric is appropriate is a common problem when it comes to correlation metrics [ Hove2018 ], including problems with Cohen’s Kappa [see Krippendorff 2004, cited over 4000 times] despite it being commonly used including many LLM as judge papers.
>
>  Hence, our recommendation in section 4.3 “Recommendations for reporting effectiveness of automated methods” - of stratification, visualization and multi-metric reporting so we can interpret the strengths and gaps in the metrics. In particular, we suggest *“2. Multi-metric reporting: If there was no uncertainty, measures such as F1 would have worked. However, as a result of uncertainty, no single metric can capture important insights about every type of data as demonstrated in Sections 3.1, 3.2 and 3.3. Thus, we recommend reporting on multiple metrics belonging to different families, such as chance and non-chance-adjusted measures, so each metric in its own way can assist in bringing the less obvious”.*
>
>
> **In terms of the use of Wasserstein Distance**, one of the challenges, is that it cannot be used for categorical values as the distance between 2 categorical values is meaningless. JSD only relies on probability of values hence can be used for categorical values as well.  However, it does not mean we cannot customize Wassterstein - it can be obviously explored and adapted further,  possibly for ordinal values where we plug bespoke distance for different points ( e.g., Likert values agree (4) – strongly agree (5), vs neutral (3) – agree (4), as distance between  2 points may not be equidistant ) . As mentioned in our paper, there is no one metric that works for all.  This highlights the central theme of the paper: no single metric can fully capture the complexities of comparing human labelers with LLM-based auto labelers.

---

> ### Author Response · Authors · 2024-11-22
> **Reviewer 4 - Q1- Q4**
>
> 1.  How do you decide what's "high" or "low" uncertainty in your stratification, and did you try other thresholds?
>
> We have now addressed this in the response to weakness 1. Let us know if you have more questions.
>
> 2. Why did you choose Jensen-Shannon Divergence for human perception, and can you show this is better than existing metrics?
>
> We have now addressed this in the response to weakness 2.
>
> 3. How did you adapt Krippendorff’s α and similar metrics to account for systematic machine errors, not just random ones?
>
>  We mention  the challenges in Section 4.1 - CHALLENGES IN METRICS AND INTERPRETABILITY --   *Errors made by LLMs are rarely predictable, yet they are not random; rather, they are reproducible, making them systematic errors. The unpredictable nature of LLMs makes it difficult to design an effective metric that compares them with humans, given the uncertainty associated with human label*.
>
> Hence as a workaround, we propose stratification, visualization  and multi-metric reporting detecting systematic errors in machine labels as mentioned in section 4.3
>
> 4. Why use previous prompts without optimizing them for each model, wouldn't this affect fairness in comparisons?
>
> As mentioned in section 3 - Analysis and settings,   we reuse the original G-Eval results (Liu et al., 2023b) which rates the quality of summaries on a scale of 1-5 and we assumed that the *original authors have optimized the results for GPT4*. We also  rely on the existing results on preference data on MT-Bench and GPT-4 from Zheng et al. (2023), that were presumably optimized for GPT-4. Our experimental results on other models further demonstrate that regardless of how / if  the prompts are optimized our central theme and findings hold  as discussed in our final section  - RECOMMENDATIONS FOR REPORTING EFFECTIVENESS OF AUTOMATIC METHODS
>
> a) Stratification by uncertainty levels: As discussed in Sec. 3.1, uncertainty in human labels can
> obfuscate performance gaps between machines and human evaluators. Hence, we strongly recom-
> mend stratifying results by uncertainty proportions.
>
> b). Multi-metric reporting: If there was no uncertainty, measures such as F1 would have worked.
> However, as a result of uncertainty, no single metric can capture important insights about every type
> of data as demonstrated in Sections 3.1, 3.2 and 3.3. Thus, we recommend reporting on multiple
> metrics belonging to different families, such as chance and non-chance-adjusted measures, so each
> metric in its own way can assist in bringing the less obvious but critical aspects about the underlying
> data to the forefront.
>
> c). Visualization of results: A single non-parametric aggregate metric can rarely capture the entirety of underlying raw data, and hence visualization is key to understanding performance gaps, as
> discussed in Section 3.3. The proposed perception charts are a step towards making aggregate cor-
> relation more interpretable, as well as highlighting the strengths and gaps of the automatic labelellers

---

> ### Author Response · Authors · 2024-11-22
> **Reviewer 4 - Q5**
>
> 1. Can you explain how you created the synthetic data for high uncertainty and whether it really reflects real-world data?
>
>
> We have also now included the code in the Appendix A.11  to clarify how the random simulated dataset  was created. We have now also referred to it in RQ1. Here is the code we use to generate the synthetic dataset.
>
> ```
> import random
> import numpy as np
> import random
> import pandas as pd
>
> def synthetic_random_nominal_dataset():
>     """
>     Simulates a random binary dataset with 2 human labellers.
>     The 2 humans can either (a) both pick 0 (b)both pick 1 (c) one picks 0 and the other picks 1 or vice versa
>     :return:
>     """
>     dataset_size = 200
>     humans_2_one_pick_0_other_picks_1 = [random.sample([0, 1], 2) for _ in range(dataset_size // 2)]
>     humans_2_both_pick_1 = [[1, 1] for _ in range(dataset_size // 4)]
>     humans_2_both_pick_0 = [[0, 0] for _ in range(dataset_size // 4)]
>
>     human_2_annotators_binary_simulated = humans_2_one_pick_0_other_picks_1 + humans_2_both_pick_1 + humans_2_both_pick_0
>
>     random_labeller_choice = [np.random.choice([1, 0]) for _ in range(dataset_size)]
>
>     # Final df
>     df = pd.DataFrame(data={"human_labels": human_2_annotators_binary_simulated,
>                             "random_labeller": random_labeller_choice
>                             })
>
>     return df
>
> def synthetic_random_ordinal_dataset():
>     """
>     Simulates a random 3 way classification 1-2-3, with 2 human labellers.
>     The 2 humans can either (a) both pick 1 (b)both pick 2. and so on (c) disagree
>     :return:
>     """
>     dataset_size = 600
>     humans_disagree = [random.sample([1, 2, 3], 2) for _ in range(dataset_size // 2)]
>     humans_agree_1 = [[1, 1] for _ in range(dataset_size // 6)]
>     humans_agree_2 = [[2, 2] for _ in range(dataset_size // 6)]
>     humans_agree_3 = [[3, 3] for _ in range(dataset_size // 6)]
>
>     human_2_annotators_ordinal_simulated = humans_disagree + humans_agree_1 + humans_agree_2 + humans_agree_3
>
>     random_labeller_choice = [np.random.choice([1, 2, 3]) for _ in range(dataset_size)]
>
>     df = pd.DataFrame(data={"human_labels": human_2_annotators_ordinal_simulated,
>                             "random_labeller": random_labeller_choice
>                             })
>
>     return df
> ```
>
>
> The aim of this dataset is to demonstrate how uncertainty can impact the results when solely relying on a single number. In addition,  we have 4  additional datasets (Topic chat, DICES and QAGS) used in popular LLM as judge papers ) as a case study in the Appendix (section A.11) of our paper and referred in the main body in section 4.3. This is in addition to existing 6 datasets (SNLI, MNLI-matched, MNLI-mismatched, SummEval, Mt-bench and synthetic datasets ) on several models – Mistral, Sonnet, LLama and GPT-4.
>
> Our findings hold across both synthetic and real world datasets -- in a stratified group with higher proportion of noisy or uncertain samples (as measured by low HH correlation), the HM correlation seems to outperform HH correlation.

---

> > ### Comment · Reviewer_RFAo · 2024-11-26
> > **Comments after rebuttal**
> >
> > Thank you for your answers and clarifications. I have updated my scores reflecting the current information.

---

> > > ### Author Response · Authors · 2024-11-26
> > > **Response to reviewer 4 RFAo**
> > >
> > > Thank you!

---

> ### Author Response · Authors · 2024-11-22
> **Reviewer 4 - Q7**
>
> 1. How would these findings help practitioners improve real-world model evaluations?
>
> We have *expanded Appendix A.11 to include a case study  with four additional datasets (Topic chat, DICES and QAGS)*, This is in addition to existing 6 datasets (SNLI, MNLI-matched, MNLI-mismatched, SummEval, Mt-bench and synthetic datasets ) on several models – Mistral, Sonnet, LLama and GPT-4 . These datasets are used in popular LLM as a judgepaper to draw conclusions of the LLM capabilities. we summaries the findings of the casestudy ( detailed numbers in the Appendix A.11)
>
> - **Effects of Multi-metric reporting:** On the Topical Chat (TC) dataset, for understandable criteria  the aggregate number (Krippendorff-$\alpha$ -0.01)  of $H^wM^w$ score of -0.01 *superficially seems to imply* that HM correlation is low as shown in Table6. However,  percentage agreement (score 0.97) and Randolph-$\kappa$ (score 0.93) score quite highly, indicating that class imbalance has substantially lowered  Krippendorff-$\alpha$ pretty close to 0.0. Also note that over 96\% of the samples have perfect human agreement, however the overall HH Krippendorff-$\alpha$ is quite low scoring -0.01. This effect of how  various chance adjusted metrics impact correlation scores is also discussed in detail in section 4.3.
>
>  - **Impact of stratification** When we compare the overall performance (column All in Table-6 on dataset TC (criteria understandable) with dataset  QAGS, Randolph-$\kappa$ drops substantially by 19 points (0.93 $\rightarrow$ 0.74). However, QAGS dataset has around 66\%  of the samples that have perfect human agreement, while TC has 96\% of the sample with human agreement.  When we compare  the samples with perfect human agreement between the 2 datasets, the model performance gap reduces to just 6 points (0.93 $\rightarrow$ 0.87), pointing to how  comparing datasets with different proportion  of uncertain samples can affect our conclusion ( in this case *incorrectly  that the model  performance is substantially lower in QAGS compared to TC*). With the DICES dataset (crowdsourced with over 100 annotations per item, with no perfect agreement items), on the other hand, the model seems to struggle across all metrics and stratification groups, indicating much deeper investigation is required.  The general trend of where in a stratified group with relatively higher proportions of  noisy or uncertain samples   (as measured by low HH correlation), the $H^wM^w$ correlation seems to outperform HH correlation as \mycolorbox{LightBlue}{highlighted}, as shown in Table 8} also applies, indicating to how \textit{models can superficially appear to approximate human majority when proportion samples of uncertainty is relatively higher} as  discussed in Section RQ1.
>
>  Our framework, by pinpointing the source of discrepancies, can guide practitioners toward addressing problems on the appropriate side and draw the right conclusion instead of inferring (sometimes  erroneously ) from a single aggregate that either models are better or that models are inadequate as demonstrated in our paper.

---

> ### Author Response · Authors · 2024-11-22
> **Response to reviewer 4 - Q6**
>
> 1. Could you provide concrete examples where machine evaluations diverge significantly from human judgments?
>
> We have now added concrete examples in the appendix, Table 11 from SNLI dataset, where humans have perfect agreement (5/5 votes for the majority label), but the  LLM predicts a different label.
> Here are some of those examples
>
>
>
> Example 1 :
>
> - Premise: A brown a dog and a black dog in the edge of the ocean with a wave under them boats are on the water in the background.
> - Hypothesis: The dogs are swimming among the boats.
> - Human Labels: entailment; entailment; entailment; entailment; entailment
> - Machine Labels: neutral; neutral; neutral; neutral; neutral
>
>
> Example 2
>
> - Premise: A young child is jumping into the arms of a woman wearing a black swimming suit while
> in a pool.
> - Hypothesis: Mother catching her son in a pool.
> - Human Labels: neutral; neutral; neutral; neutral; neutral
> - Machine Labels: entailment; entailment; entailment; entailment; entailment
>
>
> Example 3:
>
> - Premise: Two women are embracing while holding to go packages.
> - Hypothesis: The men are fighting outside a deli.
> - Human Labels: contradiction; contradiction; contradiction; contradiction; contradiction
> - Machine Labels: neutral; neutral; neutral; neutral; neutral
>
>
> Example 4
>
> - Premise: Two young children in blue jerseys, one with the number 9 and one with the number 2 are
> standing on wooden steps in a bathroom and washing their hands in a sink.
> - Hypothesis: Two kids in jackets walk to school.
> - Human Labels: contradiction; contradiction; contradiction; contradiction; contradiction
> - Machine Labels: neutral; neutral; neutral; neutral; neutral
>
>
> Example 5
>
> - Premise: Three women in dress suits walk by a building.
> - Hypothesis: Three women are traveling by foot.
> - Human Labels: entailment; entailment; entailment; entailment; entailment
> - Machine Labels: neutral; neutral; neutral; neutral; neutral

---

### Official Review · Reviewer_b4vj · 2024-11-02

**Soundness:** 3
**Presentation:** 3
**Contribution:** 3
**Rating:** 8
**Confidence:** 5

**Summary:**

This paper explores how current methods for evaluating generative models often fall short by relying too heavily on correlation metrics like Krippendorff’s α and Randolph’s κ. These metrics, while common, can mask important nuances in human judgment, especially in cases where human responses vary widely. The authors show that when there’s a high degree of variation in human evaluations, machine judgments might seem to align well, but as human consensus strengthens, this alignment breaks down, revealing gaps in machine understanding. To address these issues, the paper proposes a more robust evaluation framework that includes stratifying results by the level of human agreement and introducing a new metric, the binned Jensen-Shannon Divergence, to better capture perception-based evaluations. Additionally, the authors suggest using visual tools like perception charts to more clearly illustrate where machine judgments align or diverge from human benchmarks. By combining multiple metrics and visualization methods, this approach aims to provide a more accurate and comprehensive understanding of automated evaluations, especially in areas where human judgments are inherently uncertain.

**Strengths:**

This paper makes a valuable contribution by tackling the often-overlooked role of human uncertainty in evaluating generative models. It’s original in its approach, introducing the binned Jensen-Shannon Divergence metric to better capture the nuances of human perception and using tools like perception charts to bring new depth to evaluations. The quality of the work shows through in its thorough methodology, with experiments across multiple datasets that lend strong support to the findings. The paper is also clear, with a well-structured flow and visuals that help explain complex ideas. Most importantly, the paper has real significance: its framework could reshape how we evaluate generative models, especially in areas where human judgment isn’t always straightforward. Finally, the paper’s significance lies in its potential to reshape evaluation practices for generative models, especially in applications where human judgment is inherently subjective, such as content generation, recommendation systems, and interactive AI. By emphasizing the role of human uncertainty and offering practical tools to account for it, this work highlights a crucial aspect often ignored in model evaluation. This framework could lead to more accurate and context-sensitive evaluations, particularly for models that interact with or respond to human preferences.
This paper offers a well-supported framework that deepens our understanding of human-machine evaluations, bridging the gap between traditional metrics and the complexities of human perception. Its contributions could have a lasting impact, inspiring future research and improving evaluation standards across the field of generative modeling.

**Weaknesses:**

There are some potential for improvement in paper such as expanding on the technical implementation of the binned Jensen-Shannon Divergence metric would make it more accessible to practitioners, potentially by providing step-by-step instructions or pseudocode. Additionally, testing the framework on a broader range of generative models beyond text (such as image or audio) would demonstrate its versatility. The perception charts are helpful, but they primarily show aggregate trends, which may obscure individual item-level discrepancies; adding item-level visualizations or error bars could improve clarity. To connect more concretely with real-world applications, the paper could benefit from case studies or examples where the framework enhances specific generative tasks, such as in recommender systems. Moreover, a side-by-side comparison with existing metrics like Krippendorff’s α would better illustrate the added value of the proposed metric. Including confidence intervals or statistical significance testing could also add rigor to the findings. Finally, considering potential biases in human label uncertainty, such as cultural or contextual differences among annotators, would make the framework more robust across diverse datasets. Together, these enhancements would increase the framework’s clarity, practical utility, and adoption potential.

**Questions:**

1. Could you provide more specific implementation guidance or pseudocode for this metric, perhaps in an appendix? This would help ensure reproducibility and clarity for those looking to apply it.
2. The paper attributes label uncertainty to genuine perceptual differences, but could the authors discuss other potential sources, such as cultural or contextual biases among annotators? How might such biases affect the evaluation results, and could additional stratification methods help account for them? This would make the framework more applicable across diverse datasets and ensure its robustness in various contexts.

---

> ### Author Response · Authors · 2024-11-22
> **Response to reviewer 3 - 1-2**
>
> Thank you so much for the positive feedback of our paper
>
> 1. Could you provide more specific implementation guidance or pseudocode for this metric, perhaps in an appendix? This would help ensure reproducibility and clarity for those looking to apply it.
>
>
> Thank you for the suggestion. We are also pursuing high reproducibility and would like the community to try our metrics and visualization tools. We already updated our submission to include **python implementation** and a toy example to demonstrate the computation process in Appendix A.7. Opensource is also work in progress.  The toy example  in the appendix A.7  is as follows :
>
>
> |  Item Id  | humans    |   human_median (Bin) |   model |
> |---:|:----------|---------------:|-------------:|
> |  A | [2, 2, 3] |              2 |            3,2 |
> |  B | [1, 2, 2] |              2 |            1,1|
> |  C | [2, 3, 3] |              3 |            2,2 |
>
>
>
>
>     Values compared in Bin 2 = H(item A + item B),  M (item A + item B)
>                            = H([2, 2, 3] + [1,2,2]), M([3,2] + [1,1])
>                            = H([2,2,3,1,2,2]), M([3,2,1,1])
>
>  Comparing the probability distribution of values between H and M for Bin 2, assume Llikert scale 1- 3, where the index represents the Likert value and the index value corresponds to the probability of that value:
>
>       Bin 2 JSD(H, M) = JSD(H[1/6, 4/6, 1/6], M[2/4, 1/4, 1/4])
>
>
> Translating this to a python library call shown below, would result in score , $JSD_{b2}$ = 0.31
>
>
>       from scipy.spatial import distance
>       distance.jensenshannon([1/6, 4/6, 1/6], [2/4, 1/4, 1/4]))
>
> Similarly, for Bin 3
>
>     Values compared in Bin 3 = H(item  C), M( item C)
>                           = H([2,3,3), M([2, 2])
>
>     Bin 3 JSD( H, M)  = JSD(H[0, 1/3, 2/3], M[0, 2/2, 0])
>
> This would result in $JSD_{b3}$ = 0.56
>
>
>
>
> Total binned JSD is the weighted sum of number of samples in each bin, where $Bin_2$ contains 2 samples A and B, $Bin_3$ contains 1 sample C
>
>
> Binned JSD = 2/3  * $JSD_{b2}$ + 1/3 *  $JSD_{b3}$ = 2/3 * 0.31 + 1/3 * 0.56 =  0.39
>
>
>
>
>
> 2. The paper attributes label uncertainty to genuine perceptual differences, but could the authors discuss other potential sources, such as cultural or contextual biases among annotators? How might such biases affect the evaluation results, and could additional stratification methods help account for them? This would make the framework more applicable across diverse datasets and ensure its robustness in various contexts.
>
>
> Thank you for this insightful suggestion. We agree that cultural or contextual biases among annotators could indeed contribute to label uncertainty, in addition to perceptual differences. Unfortunately, in most datasets, detailed annotator demographic or contextual information is unavailable, limiting our ability to explicitly analyze these factors. As a result, we broadly attribute uncertainty to “perceptual differences” as a working assumption.
>
> However, we acknowledge the importance of exploring cultural and contextual biases to enhance the framework’s applicability across diverse datasets. If large-scale annotator demographic data, including labels, were available, our proposed methods could be extended for fine-grained analyses. For instance, by stratifying the data by demographic attributes or combinations such as (demographic information, human median label), we could visualize or quantify potential biases. This would allow us to assess whether machine predictions align more closely with specific cultural or demographic groups. We appreciate your suggestion and plan to explore these directions in future work.

---

> ### Author Response · Authors · 2024-11-22
> **Response to reviewer 3 - 3-4**
>
> General response to weaknesses
>
> 3. Additionally, testing the framework on a broader range of generative models beyond text (such as image or audio) would demonstrate its versatility
>
>  Thank you for the suggestion. Our work was initially motivated by the widespread use of LLMs-as-a-Judge and our observations of their associated challenges. While our experiments focus on text-based generative models, we emphasize that the proposed method is modality-agnostic. Since the method does not rely on textual features, it can be directly extended to other modality, such as images. We have also made efforts to open-source our implementation to facilitate its broader adoption and adaptation for diverse modalities.
>
> 2. The perception charts are helpful, but they primarily show aggregate trends, which may obscure individual item-level discrepancies; adding item-level visualizations or error bars could improve clarity.
>
> We appreciate the suggestion.  In section 4.2 we mention that "Effective visualization is a trade-off between plotting every single data point (too much information that is hard to synthesize) and an aggregate view (summarized view where key information might be obscured). "  Prior to proposing the perception charts,  we initially attempted to use existing BlandAltman plots (https://www.ajo.com/article/s0002-9394(08)00773-3/fulltext), (item level plot) to visualize our data. but it is very difficult to synthesize, same <x,y> values tend to overlap etc, too much information also meant we couldn't find any patterns. That lead us to create the proposed set of plots. We agree that our aggregate view may not cover all the problems, however a reasonable workaround might be plot for subset of the dataset to zoom in on the problem area and when the subset size is small enough  item level plots such as BlandAltman may become useful.
>
> 4.  To connect more concretely with real-world applications, the paper could benefit from case studies or examples where the framework enhances specific generative tasks, such as in recommender systems.
>
> We have expanded Appendix A.11 to include a case study  with four additional datasets (Topic chat, DICES and QAGS). These datasets are used in popular LLM as a judgepaper to draw conclusions of the LLM capablities. we summaries the findings of the casestudy ( detailed numbers in the Appendix A.11)
>
> - **Effects of Multi-metric reporting:** On the Topical Chat (TC) dataset, for understandable criteria  the aggregate number (Krippendorff-$\alpha$ -0.01)  of $H^wM^w$ score of -0.01 *superficially seems to imply* that HM correlation is low as shown in Table6. However,  percentage agreement (score 0.97) and Randolph-$\kappa$ (score 0.93) score quite highly, indicating that class imbalance has substantially lowered  Krippendorff-$\alpha$ pretty close to 0.0. Also note that over 96\% of the samples have perfect human agreement, however the overall HH Krippendorff-$\alpha$ is quite low scoring -0.01. This effect of how  various chance adjusted metrics impact correlation scores is also discussed in detail in section 4.3.
>
> - **Impact of stratification** When we compare the overall performance (column All in Table-6 on dataset TC (criteria understandable) with dataset  QAGS, Randolph-$\kappa$ drops substantially by 19 points (0.93 $\rightarrow$ 0.74). However, QAGS dataset has around 66\%  of the samples that have perfect human agreement, while TC has 96\% of the sample with human agreement.  When we compare  the samples with perfect human agreement between the 2 datasets, the model performance gap reduces to just 6 points (0.93 $\rightarrow$ 0.87), pointing to how  comparing datasets with different proportion  of uncertain samples can affect our conclusion ( in this case *incorrectly  that the model  performance is substantially lower in QAGS compared to TC*). With the DICES dataset (crowdsourced with over 100 annotations per item, with no perfect agreement items), on the other hand, the model seems to struggle across all metrics and stratification groups, indicating much deeper investigation is required.  The general trend of where in a stratified group with relatively higher proportions of  noisy or uncertain samples   (as measured by low HH correlation), the $H^wM^w$ correlation seems to outperform HH correlation as \mycolorbox{LightBlue}{highlighted}, as shown in Table 8} also applies, indicating to how \textit{models can superficially appear to approximate human majority when proportion samples of uncertainty is relatively higher} as  discussed in Section RQ1.
>
>  Our framework, by pinpointing the source of discrepancies, can guide practitioners toward addressing problems on the appropriate side and draw the right conclusion instead of inferring (sometimes  erroneously ) from a single aggregate that either models are better or inadequate as demonstrated in our paper.

---

> ### Author Response · Authors · 2024-11-22
> **Response to reviewer 3 : 5**
>
> 5. Moreover, a side-by-side comparison with existing metrics like Krippendorff’s α would better illustrate the added value of the proposed metric
>
> To address this suggestion, we created a toy example to demonstrate the comparative strengths of our JSD-based metric against existing metrics like Krippendorff’s α. This example, detailed in Appendix A.12 . We also include the toy example below.
>
> Consider the example in the table below. If only used a single gold human label (human_median in this case, could also be human_majority) to compute correlation, the acceptable values that humans have chosen is lost. As a result, metrics such as Krippendorff will see treat any value that is equidistant from the human single “gold” label as acceptably similar. For instance, assume that humans choose “disagree” or “neutral” (median/ majority value) (selections <2,3,3>). A good model chooses ''disagree" and a bad model chooses “agree” (completely different to human choice), because both “disagree” (Likert 2) and “agree” (Likert 4) are equidistant from the median/majority value (Likert 3- Median value), K-alpha has assigned very similar scores for the model that has assigned “disagree” (Likert 2) and the model that has assigned “agree” (Likert 4). Rank correlation metrics, in addition to their misfit in comparing item level Likert scores already discussed in the paper in section RQ2, also have a similar problem and deems the model that is poor as the better model as shown below. Our proposed approach on the other hand, assigned lower (better for JSD) scores to the better model as the "good” model assigns values that the humans have chosen, compared to the poor model that has predicted a different score altogether.
>
> | humans    |   human_median |   model_good |   model_poor |
> |:----------|---------------:|-------------:|-------------:|
> | [2, 2, 3] |              2 |            3 |            1 |
> | [1, 2, 2] |              2 |            1 |            3 |
> | [2, 3, 3] |              3 |            2 |            4 |
>
> Metric results
> | metric   |   model_good |   model_poor | Does better model score better   |
> |----------|--------------|--------------|----------------------------------|
> | Tau      |         0    |         0.82 | False                            |
> | Rho      |         0    |         0.87 | False                            |
> | K-alpha  |        -0.06 |        -0.06 | False                            |
> | JS_b     |         0.56 |         0.65 | True                             |
>
>
>  **Note** Lower JSD (more similar hence indicates better model) as JSD is a divergence metric, like distance. In addition, ranges between 0 (min) and 1 max, for log to the base 2 as indicated in explanation for equation 2.
>
> The example above also exemplifies how unless we know apriori which is a better model, it is difficult to identify advantages / shortcomings with correlation measurements including the proposed binned-jsd. Effectiveness of metrics depend on the data. We dont know for certain if a model appears to be better/worse because of gaps in metrics, creating a chicken and egg problem in measuring the effectiveness of a metric itself. Not knowing which metric is appropriate is a common problem when it comes to correlation metrics [ Hove2018 ], including problems with Cohen’s Kappa [see Krippendorff 2004, cited over 4000 times] despite it being commonly used including many LLM as judge papers.
>
>  Hence, our recommendation in section 4.3 “Recommendations for reporting effectiveness of automated methods” - of stratification, visualization and multi-metric reporting so we can interpret the strengths and gaps in the metrics. In particular, we suggest *“2. Multi-metric reporting: If there was no uncertainty, measures such as F1 would have worked. However, as a result of uncertainty, no single metric can capture important insights about every type of data as demonstrated in Sections 3.1, 3.2 and 3.3. Thus, we recommend reporting on multiple metrics belonging to different families, such as chance and non-chance-adjusted measures, so each metric in its own way can assist in bringing the less obvious”.*
>
>
>
> It is important to note that our goal is not to claim superiority over existing metrics but to offer a holistic perspective and a set of tools to analyze discrepancies between human and machine labels comprehensively.

---

> ### Author Response · Authors · 2024-11-22
> **Response to reviewer 3 -6**
>
> 6 Confidence intervals and statistical significance:
>
> We absolutely agree this is a critical component that is missing on most studies, including ours. Some of the challenges with estimating variance in correlation problems is briefly described in Deutsch et al., 2021. The key problem with statistical significance is how "random chance agreement" is computed.    In our paper, we have explicitly called this out in section 4.2 of our paper as follows *"In addition, statistical analysis, such as null-hypothesis and significance testing, is essential for determining whether one model outperforms another by random chance. Here, the chance component  includes 2 aspects, **{(1)}** chance due to the  nature of samples in the evaluation set **{(2)}** uncertainty in human labels. A third aspect, even harder,  is estimating the error rate as a result of systematically unpredictable erroneous labels from any automated evaluator.
> Future studies should explore these problems, including approaches like resampling (Deutsch et al., 2021). Incorporation of chance in rank correlation is also an important aspect to account for when two models differ in rank, but the corresponding difference in  absolute scores is negligible, then the  difference in the rank may not be meaningful.*
>
> We hope to explore this in future work.

---

### Official Review · Reviewer_ApGo · 2024-11-03

**Soundness:** 2
**Presentation:** 2
**Contribution:** 2
**Rating:** 6
**Confidence:** 3

**Summary:**

The paper describes the analysis of different measures in  evaluating  LLM responses.
A measure, specifically a binned Jensen- Shannon divergence is proposed.

This measure for ordinal perception data is justified by the author since the evaluation does not need a single gold label and the human and the machine are not interchangeable. This last condition breaks a necessary condition for the Krippendorff-alpha coefficient.

A distinction is done among Nominal values, ordinal values and continuous values. The Questions are not equally proposed for the three types of data, raising some difficulty in reading the paper.

RQ1:How does uncertainty in human labels impact correlation metrics when
we measure the efficacy of automatic evaluation methods? (Sec. 3.1)

The authors state that the uncertainty in human labels is high the human-machine majority labels are similar. The meaning appears to be that if there is no concordance among labeller, in this case the LLM judge is ok. It is the LLM-judge just adding noise to the labelling process?


RQ2: How can we measure human-to-machine(HM) agreement that accounts for human uncertainty as a result of variation in human perception?
Human to machine agreement is a measure of uncertainty in human perception. For this question, the comparison with different agreement percentage is tested. For this task is proposed the binned Jensed inequality.


RQ3: How can we visualize the underlying data to draw meaningful insights when we compare the results from automatic and human evaluations?

The authors compare ordinal and perceptual based ratings between human and machines

**Strengths:**

The topic is new, and since human-labelled data are difficult to retrieve, an evaluation of machine-labelled data is very interesting.

The tests have been done on different LLMs

Multiple tests have been performed.

**Weaknesses:**

The work's presentation is not clear. The research questions help to  interpret the experiments but not all the results are clear.

Some terms, like H^W R^W (^ indicates apex), are defined in the caption of caption 1. Probably they should be defined in the text and used in the table.

It is unclear how the partitions are decided. In the experiments:in table 1 the thresholds are  0, 0.8, 1; in table 2 the threhsolds are 0.6 and 1; in table 3 are 0.6, 0.8, 1.0. It is not clear if the thresholds are experiment dependent or there is a rationale behind the threshold selection.

The random classifier test, reported in table 1 should be better described. The table reports unique =2 or unique =3 with a percentage. Unique term is not present in the description and should be explained for the clear presentation of the experiment. If the MNLI and SNLi
dataset have only a limited set of labels it should be specified in the description.

In Table 2 the terms H^mu M^mu are not specified.


In Figure 3 are shown the human perception vs the machine labels binned by human median rating. The JS value is reported.
The highest values of JS are for \bar{H}=1 and  \bar{H}=5. Looking at the histograms, the histograms of human and machine with \bar{H}=3 (and  in some measure \bar{H}=4) are very similar, but the JS values are sensibly lower.
The authors state that humans tend to be more certain when they assign extreme rating and the machine rarely provide extreme values.
Could  the explanation of the experiment take into account also this aspect? If there is a different interpretation of this discrepancy ( similar histograms but lower JS value) it would be useful to make this point more evident.

**Questions:**

Could the author provide some detail in the selection of different thresholds across the tables? It would be useful to clarify whether these thresholds are dataset-specific or if there's a general principle behind the thresholds selection.

In the paper, different metrics  are used and the human and machine labels are compared with average, with median or with majority labels. Are all these comparisons needed? Do they capture multiple aspects of the outputs?

Is the binned JSD the best metric for the proposed experiments? Is it possible to calculate this metric, or its adaptation,  for all the experiments proposed in the paper?

The bins are used to mimic human perception. Beyond the aggregation of perception, can they capture variation in human perception?

---

> ### Author Response · Authors · 2024-11-21
> **Response to reviewer 2 - Q1-3**
>
> 1. Some terms, like H^W R^W (^ indicates apex), are defined in the caption of caption 1. Probably they should be defined in the text and used in the table.
>
> Thank you for pointing this out. We have updated the text in RQ1 to indicate it as follows -- “At surface level, HM correlation seems to improve with human certainty, as shown in Table 1 – column $H^wM^w$ comparing Human majority ($H^w$) with machine majority ($M^w$).”
>
>
> 2. It is unclear how the partitions are decided. In the experiments:in table 1 the thresholds are 0, 0.8, 1; in table 2 the threhsolds are 0.6 and 1; in table 3 are 0.6, 0.8, 1.0. It is not clear if the thresholds are experiment dependent or there is a rationale behind the threshold selection.
>
>  In Table 1, the MNLI and SNLI datasets have exactly 5 human labels per item. Hence, you have the following scenarios – 5/5 = 1.0, 4/5 = 0.8, 3/5=0.6. Hence, we have stratified by all possible votes for the majority label.  In Table 2, for SummEval, there are exactly 3 human labels. Hence, you have 3/3=1.0, 2/3=0.67, 1/3 =0.33, for median label.   In Table 2, we only showed the max agreement and least agreement for brevity, we have now included all the stratification in Appendix In Table 5 and explained the stratification. The results do not change and remain the same.
>
>  For Table 3, Each item  has 3-5 human label (varying number of human labels), where majority of the items have closer to 3 labels. The main challenge with the MTBench dataset displayed in table 2, is the sample size is quite small for each model pair with multiple annotations. hence some partitions have less than 5 samples or even 0, unlike the MNLI or SNLI datasets which have a few thousand samples, Hence, we have reported the partitions with resonable number of samples, which is 100\%, 60-80. We have now included all the partitions with even 1  sample in Appendix in Table 6. Our  findings hold, across all partitions, in table 6 as well .
>
> 3. The random classifier test, reported in table 1 should be better described. The table reports unique =2 or unique =3 with a percentage. Unique term is not present in the description and should be explained for the clear presentation of the experiment. If the MNLI and SNLi dataset have only a limited set of labels it should be specified in the description.
>
>
>
> Thank you for your comment.  We mention in RQ1 “We also stratify samples by the number of unique human labels to ensure that our findings are consistent regardless of the stratification method.” .
> We have also now included the code in the Appendix A.11  to clarify how the random simulated dataset  was created. We have now also referred to it in RQ1. Unique here refers to the number of unique labels that human labellers assign to a given item. Here is the code we use to generate the synthetic dataset.
>
> ```
> import random
> import numpy as np
> import random
> import pandas as pd
>
> def synthetic_random_nominal_dataset():
>     """
>     Simulates a random binary dataset with 2 human labellers.
>     The 2 humans can either (a) both pick 0 (b)both pick 1 (c) one picks 0 and the other picks 1 or vice versa
>     :return:
>     """
>     dataset_size = 200
>     humans_2_one_pick_0_other_picks_1 = [random.sample([0, 1], 2) for _ in range(dataset_size // 2)]
>     humans_2_both_pick_1 = [[1, 1] for _ in range(dataset_size // 4)]
>     humans_2_both_pick_0 = [[0, 0] for _ in range(dataset_size // 4)]
>
>     human_2_annotators_binary_simulated = humans_2_one_pick_0_other_picks_1 + humans_2_both_pick_1 + humans_2_both_pick_0
>
>     random_labeller_choice = [np.random.choice([1, 0]) for _ in range(dataset_size)]
>
>     # Final df
>     df = pd.DataFrame(data={"human_labels": human_2_annotators_binary_simulated,
>                             "random_labeller": random_labeller_choice
>                             })
>
>     return df
>
> def synthetic_random_ordinal_dataset():
>     """
>     Simulates a random 3 way classification 1-2-3, with 2 human labellers.
>     The 2 humans can either (a) both pick 1 (b)both pick 2. and so on (c) disagree
>     :return:
>     """
>     dataset_size = 600
>     humans_disagree = [random.sample([1, 2, 3], 2) for _ in range(dataset_size // 2)]
>     humans_agree_1 = [[1, 1] for _ in range(dataset_size // 6)]
>     humans_agree_2 = [[2, 2] for _ in range(dataset_size // 6)]
>     humans_agree_3 = [[3, 3] for _ in range(dataset_size // 6)]
>
>     human_2_annotators_ordinal_simulated = humans_disagree + humans_agree_1 + humans_agree_2 + humans_agree_3
>
>     random_labeller_choice = [np.random.choice([1, 2, 3]) for _ in range(dataset_size)]
>
>     df = pd.DataFrame(data={"human_labels": human_2_annotators_ordinal_simulated,
>                             "random_labeller": random_labeller_choice
>                             })
>
>     return df
> ```

---

> > ### Author Response · Authors · 2024-11-21
> > **Response to reviewer 2 - Q4-7**
> >
> > 4. In Table 2 the terms H^mu M^mu are not specified.
> >
> > Thank you for pointing this out. we have now updated it, under Table 2 as follows Spearman's-$\rho$ median $(\overline{H}\overline{M}$) vs. mean ($H^{\mu}M^{\mu}$).
> >
> > 5. In Figure 3 are shown the human perception vs the machine labels binned by human median rating. The JS value is reported. The highest values of JS are for \bar{H}=1 and \bar{H}=5. Looking at the histograms, the histograms of human and machine with \bar{H}=3 (and in some measure \bar{H}=4) are very similar, but the JS values are sensibly lower. The authors state that humans tend to be more certain when they assign extreme rating and the machine rarely provide extreme values. Could the explanation of the experiment take into account also this aspect? If there is a different interpretation of this discrepancy ( similar histograms but lower JS value) it would be useful to make this point more evident.
> >
> >
> >
> > Binned-JSD, is based on JSD and therefore is a divergence metric (like distance). Hence, lower scores indicate smaller distance implying more similar distributions.  So lower is better, when comparing humans and machines. We have also mentioned this in the paper, under equation 2 in RQ2 - “Since JSD is a distance-based measure, lower scores are better because they indicate that the human and machine judgments are similar.”
> >
> >
> > 6. Could the author provide some detail in the selection of different thresholds across the tables? It would be useful to clarify whether these thresholds are dataset-specific or if there's a general principle behind the thresholds selection.
> >
> > Yes- Thank you for your suggestion to improve the clarity of the paper. We have now  addressed this as part of your question 2
> >
> > 7.  In the paper, different metrics are used and the human and machine labels are compared with average, with median or with majority labels. Are all these comparisons needed? Do they capture multiple aspects of the outputs?
> >
> > We have used majority label for categorical values where there is no natural order ( such as preferencing model A vs Model B or fact checking ). Median values are typically applied for ordinal values ( statistically speaking), although  many research papers have been using mean to aggregate likert values, and we have detailed this in section RQ2 -   *“Furthermore, whether to treat Likert-data as ordinal or interval data dictates the aggregation method – median or mean, is also debated (Joshi et al., 2015). The argument against using Likert-data as interval data is that the points on the scale may not be equidistant, e.g., the distance between pair ⟨neutral, agree⟩ may not be the same as the distance between ⟨agree, strongly agree⟩. We report Spearman’s-ρ for both to illustrate the difference in Table 2.”*

---

> ### Author Response · Authors · 2024-11-21
> **Response to reviewer 2 - Q 8**
>
> 8. Is the binned JSD the best metric for the proposed experiments? Is it possible to calculate this metric, or its adaptation, for all the experiments proposed in the paper?
>
> Reviewer 1 also has asked a similar question. Here is our explanation
>
> The use of binned-JSD solves a specific problem, where a single majority label is not sufficient to represent the human preference as mentioned in section 4.2.
>
> The advantage of binned-JSD  can be demonstrated  using a toy example (also now added to the appendix A.12 and referred in the paper in section RQ2) as follows.  if only used a single gold human label (human_median in this case, could also be human_majority) to compute correlation, the acceptable values that humans have chosen is lost. As a result, metrics such as Krippendorff will see treat any value that is equidistant from the human single “gold” label as acceptably similar. For instance, assume that humans choose “disagree” or “neutral” (median/ majority value) (selections <2,3,3>). A good model chooses ''disagree" and a bad model chooses “agree” (completely different to human choice), because both “disagree” (Likert 2) and “agree” (Likert 4) are equidistant from the median/majority value (Likert 3- Median value), K-alpha has assigned very similar scores for the model that has assigned “disagree” (Likert 2) and the model that has assigned “agree” (Likert 4). Rank correlation metrics, in addition to their misfit in comparing item level Likert scores already discussed in the paper in section RQ2, also have a similar problem and deems the model that is poor as the better model as shown below. Our proposed approach on the other hand, assigned lower (better for JSD) scores to the better model as the "good” model assigns values that the humans have chosen, compared to the poor model that has predicted a different score altogether.
>
> | humans    |   human_median |   model_good |   model_poor |
> |:----------|---------------:|-------------:|-------------:|
> | [2, 2, 3] |              2 |            3 |            1 |
> | [1, 2, 2] |              2 |            1 |            3 |
> | [2, 3, 3] |              3 |            2 |            4 |
>
> Metric results
> | metric   |   model_good |   model_poor | Does better model score better   |
> |----------|--------------|--------------|----------------------------------|
> | Tau      |         0    |         0.82 | False                            |
> | Rho      |         0    |         0.87 | False                            |
> | K-alpha  |        -0.06 |        -0.06 | False                            |
> | JS_b     |         0.56 |         0.65 | True                             |
>
>
>
> The example above also exemplifies how unless we know apriori which is a better model, it is difficult to identify advantages / shortcomings with correlation measurements including the proposed binned-jsd. Effectiveness of metrics depend on the data. We dont know for certain if a model appears to be better/worse because of gaps in metrics, creating a chicken and egg problem in measuring the effectiveness of a metric itself. Not knowing which metric is appropriate is a common problem when it comes to correlation metrics [ Hove2018 ], including problems with Cohen’s Kappa [see Krippendorff 2004, cited over 4000 times] despite it being commonly used including many LLM as judge papers.
>
>  Hence, our recommendation in section 4.3 “Recommendations for reporting effectiveness of automated methods” - of stratification, visualization and multi-metric reporting so we can interpret the strengths and gaps in the metrics. In particular, we suggest *“2. Multi-metric reporting: If there was no uncertainty, measures such as F1 would have worked. However, as a result of uncertainty, no single metric can capture important insights about every type of data as demonstrated in Sections 3.1, 3.2 and 3.3. Thus, we recommend reporting on multiple metrics belonging to different families, such as chance and non-chance-adjusted measures, so each metric in its own way can assist in bringing the less obvious”.*
>
> Through this paper and the arguments we make, we would like to encourage the  research community  to take a deeper look at metrics to understand the gaps between metrics vs the reality of comparing machine with human judgements as a result of uncertainty.

---

> ### Author Response · Authors · 2024-11-21
> **Response to reviewer 2 - Q 9**
>
> 9 . The bins are used to mimic human perception. Beyond the aggregation of perception, can they capture variation in human perception?
> Yes, bins capture variation in human perception. Reviewer 1 has also asked a very similar question.
>
> Within each bin, the JSD captures the difference between the distribution human and machine labels. For example, if the human label distribution spreads over multiple labels in a bin (high variation), while the machine label distribution concentrates on one label, then the JSD metric would capture that. Here is a example,
>
>
> |  Item Id  | humans    |   human_median (Bin) |   model |
> |---:|:----------|---------------:|-------------:|
> |  A | [2, 2, 3] |              2 |            3,2 |
> |  B | [1, 2, 2] |              2 |            1,1|
> |  C | [2, 3, 3] |              3 |            2,2 |
>
>
>
>
>     Values compared in Bin 2 = H(item A + item B),  M (item A + item B)
>                            = H([2, 2, 3] + [1,2,2]), M([3,2] + [1,1])
>                            = H([2,2,3,1,2,2]), M([3,2,1,1])
>
>  Comparing the probability distribution of values between H and M for Bin 2, assume Llikert scale 1- 3, where the index represents the Likert value and the index value corresponds to the probability of that value:
>
>       Bin 2 JSD(H, M) = JSD(H[1/6, 4/6, 1/6], M[2/4, 1/4, 1/4])
>
>
> Translating this to a python library call shown below, would result in score , $JSD_{b2}$ = 0.31
>
>
>       from scipy.spatial import distance
>       distance.jensenshannon([1/6, 4/6, 1/6], [2/4, 1/4, 1/4]))
>
> Similarly, for Bin 3
>
>     Values compared in Bin 3 = H(item  C), M( item C)
>                           = H([2,3,3), M([2, 2])
>
>     Bin 3 JSD( H, M)  = JSD(H[0, 1/3, 2/3], M[0, 2/2, 0])
>
> This would result in $JSD_{b3}$ = 0.56
>
>
>
>
> Total binned JSD is the weighted sum of number of samples in each bin, where $Bin_2$ contains 2 samples A and B, $Bin_3$ contains 1 sample C
>
>
> Binned JSD = 2/3  * $JSD_{b2}$ + 1/3 *  $JSD_{b3}$ = 2/3 * 0.31 + 1/3 * 0.56 =  0.39
>
>
> We have now included these examples in the Appendix A.7 along with the full code.

---

> > ### Author Response · Authors · 2024-11-22
> > **Response to Reviewer 2**
> >
> > Lets us know if you have any further questions. Thank you for reviewing our paper !

---

> > > ### Comment · Reviewer_ApGo · 2024-11-25
> > >
> > > Thank you for your replies and updates! I have adjusted my review.

---

> ### Author Response · Authors · 2024-11-25
> **Reviewer 2**
>
> Thanks for responding to our comment. The scores does  not seem to have been updated, is there any specific items you would like us to elaborate on.   Thank you for taking the time to review our paper and help improve it :-)

---

> > ### Author Response · Authors · 2024-11-27
> > **Reviewer 2 APGo Question**
> >
> > Dear reviewer,
> > Following up to check, since you mentioned that you have adjusted the review, the scores have not been updated.   When you get a chance, it would be great if you could confirm - Thank you :-)

---

> > > ### Comment · Reviewer_ApGo · 2024-11-27
> > >
> > > The update was not saved, it should be ok now.

---

### Official Review · Reviewer_Y99p · 2024-11-04

**Soundness:** 3
**Presentation:** 4
**Contribution:** 3
**Rating:** 8
**Confidence:** 3

**Summary:**

The paper discusses the current landscape of using LLMs as judges for various tasks and presents compelling arguments for why existing correlation metrics might not take into account variations and uncertainty in human judgment.

After the author's reply (which includes more datasets and clarifications about the metric), I believe that the paper makes good contributions and good cases of showing how to be careful when using LLMs as judges. I there recommend to accept the paper!

**Strengths:**

1) The paper is well structured and presented and is very clear and easy to follow and read.
2) The paper clearly shows the issue related to relying on a high correlation between human and machine-generated outputs, giving cases where the uncertainty in human annotations is high; these correlations seem to be high, but that could also be the case even when the labeling is random.
3) The study proposed a metric, namely binned JSD, to account for variations and uncertainty in human judgment.

**Weaknesses:**

1) It is not clear to me how this new metric handles the issues raised with traditional metrics. Could the author clarify and show cases of how JSD improves the analysis of LLMs as judges when compared to human judgment? It seems like a promising direction, but I am not convinced due to the limited number of datasets and support of the authors' claim.

2) The type of human annotation is limited. Although the paper presented very well the type of collected human annotations, I limited these in their evaluations. Given the space and the breed of the paper, I suggest adding a few more datasets (Please check paper [1] for some guidance on what dataset to choose.)

3) The study considered four datasets with variant tasks and annotations that have sufficient human annotators. I think the number of annotations was well considered, but the number of the dataset could be improved, and metal analysis could have been better presented instead of large tables per dataset.


[1] Bavaresco, Anna, et al. "Llms instead of human judges? a large scale empirical study across 20 nlp evaluation tasks." arXiv preprint arXiv:2406.18403 (2024).

**Questions:**

How come the metric does not need a single value to approximate human labels but relies on a single "human and machine labels are not treated interchangeably, as the items in a given bin are selected by the human median or majority value"? This seems contradictory to me.

---

> ### Author Response · Authors · 2024-11-19
> **Response to reviewer 1 -  question 1**
>
> 1.  It is not clear to me how this new metric handles the issues raised with traditional metrics. Could the author clarify and show cases of how JSD improves the analysis of LLMs as judges when compared to human judgment? It seems like a promising direction, but I am not convinced due to the limited number of datasets and support of the authors' claim.
>
> Thank you for your comments to improve the clarity of the paper. The use of binned-JSD solves a specific problem, where a single majority label is not sufficient to represent the human preference as mentioned in section 4.2.
>
> The advantage of binned-JSD  can be demonstrated  using a toy example (also now added to the appendix A.12 and referred in the paper in section RQ2) as follows.  if only used a single gold human label (human_median in this case, could also be human_majority) to compute correlation, the acceptable values that humans have chosen is lost. As a result, metrics such as Krippendorff will see treat any value that is equidistant from the human single “gold” label as acceptably similar. For instance, assume that humans choose “disagree” or “neutral” (median/ majority value) (selections <2,3,3>). A good model chooses ''disagree" and a bad model chooses “agree” (completely different to human choice), because both “disagree” (Likert 2) and “agree” (Likert 4) are equidistant from the median/majority value (Likert 3- Median value), K-alpha has assigned very similar scores for the model that has assigned “disagree” (Likert 2) and the model that has assigned “agree” (Likert 4). Rank correlation metrics, in addition to their misfit in comparing item level Likert scores already discussed in the paper in section RQ2, also have a similar problem and deems the model that is poor as the better model as shown below. Our proposed approach on the other hand, assigned lower (better for JSD) scores to the better model as the "good” model assigns values that the humans have chosen, compared to the poor model that has predicted a different score altogether.
>
> | humans    |   human_median |   model_good |   model_poor |
> |:----------|---------------:|-------------:|-------------:|
> | [2, 2, 3] |              2 |            3 |            1 |
> | [1, 2, 2] |              2 |            1 |            3 |
> | [2, 3, 3] |              3 |            2 |            4 |
>
> Metric results
> | metric   |   model_good |   model_poor | Does better model score better   |
> |----------|--------------|--------------|----------------------------------|
> | Tau      |         0    |         0.82 | False                            |
> | Rho      |         0    |         0.87 | False                            |
> | K-alpha  |        -0.06 |        -0.06 | False                            |
> | JS_b     |         0.56 |         0.65 | True                             |
>
>
>
> The example above also exemplifies how unless we know apriori which is a better model, it is difficult to identify advantages / shortcomings with correlation measurements including the proposed binned-jsd. Effectiveness of metrics depend on the data. We dont know for certain if a model appears to be better/worse because of gaps in metrics, creating a chicken and egg problem in measuring the effectiveness of a metric itself. Not knowing which metric is appropriate is a common problem when it comes to correlation metrics [ Hove2018 ], including problems with Cohen’s Kappa [see Krippendorff 2004, cited over 4000 times] despite it being commonly used including many LLM as judge papers.
>
>  Hence, our recommendation in section 4.3 “Recommendations for reporting effectiveness of automated methods” - of stratification, visualization and multi-metric reporting so we can interpret the strengths and gaps in the metrics. In particular, we suggest *“2. Multi-metric reporting: If there was no uncertainty, measures such as F1 would have worked. However, as a result of uncertainty, no single metric can capture important insights about every type of data as demonstrated in Sections 3.1, 3.2 and 3.3. Thus, we recommend reporting on multiple metrics belonging to different families, such as chance and non-chance-adjusted measures, so each metric in its own way can assist in bringing the less obvious”.*
>
> Through this paper and the arguments we make, we would like to encourage the  research community  to take a deeper look at metrics to understand the gaps between metrics vs the reality of comparing machine with human judgements as a result of uncertainty.
>
>
>
>     [1] Klaus Krippendorff, Reliability in Content Analysis: Some Common Misconceptions and Recommendations, Human Communication Research, Volume 30, Issue 3, July 2004, Pages 411–433, https://doi.org/10.1111/j.1468-2958.2004.tb00738.x
>
>     [2] Debby ten Hove, Terrence D. Jorgensen, and L. Andriesvan der Ark. 2018. On the usefulness of interrater reliability coefficients. In Quantitative Psychology, pages 67–75, Cham. Springer International Publishing

---

> ### Author Response · Authors · 2024-11-19
> **Response to reviewer 1 - question 2 - 4**
>
> 2. The type of human annotation is limited. Although the paper presented very well the type of collected human annotations, I limited these in their evaluations. Given the space and the breed of the paper, I suggest adding a few more datasets (Please check paper [1] for some guidance on what dataset to choose.)
>
> We have now included 3 additional datasets (Topic chat, DICES and QAGS) used in paper[1]) as a case study  in the Appendix  (section A.11) of our paper and referred  in the main body in section 4.3. This is in addition  to existing 6 datasets (SNLI, MNLI-matched, MNLI-mismatched, SummEval, Mt-bench and synthetic datasets ) on several models – Mistral, Sonnet, LLama and GPT-4.  Our findings hold, and we summarize the findings here for the new 3 datasets.
>
> - **Effects of Multi-metric reporting:** On the Topical Chat (TC) dataset, for understandable criteria  the aggregate number (Krippendorff-$\alpha$ -0.01)  of $H^wM^w$ score of -0.01 *superficially seems to imply* that HM correlation is low as shown in Table6. However,  percentage agreement (score 0.97) and Randolph-$\kappa$ (score 0.93) score quite highly, indicating that class imbalance has substantially lowered  Krippendorff-$\alpha$ pretty close to 0.0. Also note that over 96\% of the samples have perfect human agreement, however the overall HH Krippendorff-$\alpha$ is quite low scoring -0.01. This effect of how  various chance adjusted metrics impact correlation scores is also discussed in detail in section 4.3.
>
>  - **Impact of stratification** When we compare the overall performance (column All in Table-6 on dataset TC (criteria understandable) with dataset  QAGS, Randolph-$\kappa$ drops substantially by 19 points (0.93 $\rightarrow$ 0.74). However, QAGS dataset has around 66\%  of the samples that have perfect human agreement, while TC has 96\% of the sample with human agreement.  When we compare  the samples with perfect human agreement between the 2 datasets, the model performance gap reduces to just 6 points (0.93 $\rightarrow$ 0.87).The model seems to struggle with DICES dataset (crowdsourced with over 100 annotations per item, with no perfect agreement items) across all metrics and stratification groups, indicating much deeper investigation is required.  The general trend of where in a stratified group with higher proportion of  noisy or uncertain samples   (as measured by low HH correlation), the HM correlation seems to outperform HH correlation  in Table6  applies as previously discussed in Section RQ1.
>
> 3.  The study considered four datasets with variant tasks and annotations that have sufficient human annotators. I think the number of annotations was well considered, but the number of the dataset could be improved, and metal analysis could have been better presented instead of large tables per dataset.
>
> We have increased the datasets as suggested. We acknowledge that the information is complex to synthesize as the table present results stratified by different groups and metrics. This was the simplest way with  our best efforts. Happy to incorporate any specific suggestions you have.
>
>
> 4. How come the metric does not need a single value to approximate human labels but relies on a single "human and machine labels are not treated interchangeably, as the items in a given bin are selected by the human median or majority value"? This seems contradictory to me.
>
> Binned-jsd considers the variation of the human labels. The bin an item belongs is assigned by median or majority value and hence is only used to assign the bin numbers. In the example below, we compare probability distribution of the human labels in bin 2 [2,2,3,1,2,2] with corresponding machine labels model [3,3,1,1]. We repeat the same for each bin, in this example bin 3 as well, and do a weighted sum of JSD for each bin, so that the binned-JSD always ranges between [0, 1].
>
> | humans    |   human_median (bin) |   model |
> |:----------|---------------:|-------------:|
> | [2, 2, 3] |              2 |            [3,3] |
> | [1, 2, 2] |              2 |            [1,1] |
> | [2, 3, 3] |              3 |            [2,1] |

---

> > ### Comment · Reviewer_Y99p · 2024-11-19
> > **Further clarification**
> >
> > Thanks a lot for your extensive reply, adding more datasets, and clarifying the metric details. I have two more clarifications (hopefully the latest):
> >
> > In your second table, the column titled "Does better model score better" shows that it is True for JS_b. But its value is lower for good_model. Should this also be False? Did I miss something in your reply?
> >
> > I will be happy if you also clarify the last example. It seems to me that bin 2 has no correspondence in the table.
> >
> > These toy examples are good for illustrating the metric and the problems occurring in the literature. It would be super good to include them in the paper.
> >
> > I am mostly happy with your reply and the extensive effort you made. I will most probably increase my ratings.
> >
> > Thanks!

---

> ### Author Response · Authors · 2024-11-20
> **Response to further clarification**
>
> Thank you so much for the positive feedback.
>
> 1. In your second table, the column titled "Does better model score better" shows that it is True for JS_b. But its value is lower for good_model. Should this also be False?
>
>
> Binned-JSD,  being based on JSD,   is a divergence metric (like distance), with 0 smallest ''distance'' and 1 being the maximum. Hence, lower scores indicate smaller distance, implying more similar distributions. When comparing humans and machines, lower $JS_b$ score is better,  indicating more similar distribution. We have now emphasized this in the paper, under equation 2 in RQ2 -“Since JSD is like a distance-based measure, lower scores are better because they indicate that the human and machine judgments are similar.
>
> 2. Clarify the last example. It seems to me that bin 2 has no correspondence in the table.
>
>
> |  Item Id  | humans    |   human_median (Bin) |   model |
> |---:|:----------|---------------:|-------------:|
> |  A | [2, 2, 3] |              2 |            3,2 |
> |  B | [1, 2, 2] |              2 |            1,1|
> |  C | [2, 3, 3] |              3 |            2,2 |
>
>
>
>
>     Values compared in Bin 2 = H(item A + item B),  M (item A + item B)
>                            = H([2, 2, 3] + [1,2,2]), M([3,2] + [1,1])
>                            = H([2,2,3,1,2,2]), M([3,2,1,1])
>
>  Comparing the probability distribution of values between H and M for Bin 2, assume Llikert scale 1- 3, where the index represents the Likert value and the index value corresponds to the probability of that value:
>
>       Bin 2 JSD(H, M) = JSD(H[1/6, 4/6, 1/6], M[2/4, 1/4, 1/4])
>
>
> Translating this to a python library call shown below, would result in score , $JSD_{b2}$ = 0.31
>
>
>       from scipy.spatial import distance
>       distance.jensenshannon([1/6, 4/6, 1/6], [2/4, 1/4, 1/4]))
>
> Similarly, for Bin 3
>
>     Values compared in Bin 3 = H(item  C), M( item C)
>                           = H([2,3,3), M([2, 2])
>
>     Bin 3 JSD( H, M)  = JSD(H[0, 1/3, 2/3], M[0, 2/2, 0])
>
> This would result in $JSD_{b3}$ = 0.56
>
>
>
>
> Total binned JSD is the weighted sum of number of samples in each bin, where $Bin_2$ contains 2 samples A and B, $Bin_3$ contains 1 sample C
>
>
> Binned JSD = 2/3  * $JSD_{b2}$ + 1/3 *  $JSD_{b3}$ = 2/3 * 0.31 + 1/3 * 0.56 =  0.39
>
>
> We have now included these examples in the Appendix A.7 along with the full code.

---

> > ### Comment · Reviewer_Y99p · 2024-11-20
> >
> > Thank you! I have adjusted my review and recommendation!

---

> > > ### Author Response · Authors · 2024-11-20
> > > **Thank you**
> > >
> > > Thank you so much!

---

### Author Response · Authors · 2024-12-03
**Rebuttal summary**

We would like to thank the reviewers for providing us with positive reviews and valuable feedback to enhance the clarity of our paper.

Our key objectives  of this paper have been to highlight

1. How **aggregate correlation scores  can misguide researchers** into concluding that LLM-as-Judge approximates human majority where human majority-machine correlation scores can seem higher than human-human correlation. In this paper, we show over multiple datasets that this is easier to achieve when the proportion of samples with human label uncertainty is quite high. As the proportion of samples with consistent human labels increase, this assumption that LLM-as-Judge approximates human majority falls through. In addition to using benchmark datasets, we also use a synthetic dataset to show how under high uncertainty, even a random labeller can appear to approximate human majority. Hence, our main takeaway  is that  human label uncertainty cannot be overlooked when measuring LLM-as-judge capabilities.

2. This finding led us to the follow-on question on **how to measure  LLM-as-judge capabilities, when the task is inherently subjective** and humans are likely to vary and consistent labels is not practically possible as there is no single ground truth / gold answer. To mitigate this, we propose a binned-JSD as a step in this direction to compare human labels with machine labels without assuming that a single gold label captures human choice.

3. A third contribution of this paper, is highlighting how all correlation metrics rely on some key assumptions about the nature of the human labellers and that no single metric is a perfect metric, including the proposed JSD.   This problem exacerbates with LLMs, as their nature is unpredictable.  Each metric has its own strength and weaknesses and highlights certain aspects of the underlying data. Hence, we **recommend that  researchers stratify the results by uncertainty, perform multi metric reporting and visualize the results to interpret aggregate numbers appropriately**. The proposed perception charts are a step in this direction to visualize notoriously challenging correlation numbers.

The key themes of the feedback, now addressed,  we received from our reviewers  has been

1. **The value of the proposed binned-JSD is not clear** - We have now illustrated this using a toy example in Appendix A.12. We have also included a python implementation of binned-JSD for reproducibility in Appendix A.7.

2. **How practitioners can use our framework  in real world scenarios is not clear** - We have included a case study (Appendix A.11) using 4 additional  datasets (to add to the existing 5 datasets included in the initial version of the paper)  used in popular papers to highlight how the conclusion that the machine is better or worse can be misleading as a result of either the human uncertainty or metric unsuitability.

3. **Implementation details not clear - The stratification threshold and  how the synthetic data with random labeller was created** - We have now included details of how we stratify by all possible number of  votes that a majority or median label  (e.g. 5/5, 4/5,3/5 ..)  can receive and along with results across all partitions in  Appendix In Table 5 and Table 6. We have included a python implementation of how the synthetic dataset was created in  Appendix A10.

---

### Meta-Review · Area_Chair_5hZH · 2024-12-20

**Metareview:**

This paper addresses the challenges of using aggregate correlation scores to evaluate the performance of LLMs as judges in subjective tasks. The authors argue that high human label uncertainty can misleadingly make LLMs appear to align closely with human majority labels, even more than human-human agreement. The experiments based on both benchmark and synthetic datasets, aim to show that as human label consistency increases, this assumption breaks down, emphasizing the importance of accounting for label uncertainty. The authors also introduce a binned JSD metric as an alternative and make additional suggestions  to use multi-metric reporting and stratification.

The reviewers are mostly positive, only one reviewer rates the paper marginally below acceptance threshold. Most of the criticism is around value of the binned-jsd metric and implementation / how a practitioner would use the methodology.

My recommendation is based on reviewer feedback which is mostly positive. The one critical reviewer does not engage much beyond the review and acknowledging the rebuttal.

**Additional Comments On Reviewer Discussion:**

The authors respond to the reviews in a detailed manner and it most of the reviewers seem satisfied with their response.

---

### Decision · Program_Chairs · 2025-01-22

Accept (Poster)